# Early-adult methionine restriction reduces methionine sulfoxide and extends lifespan in *Drosophila*

Hina Kosakamoto [1,2,5], Fumiaki Obata [1,2,3,5] ✉, Junpei Kuraishi[1], Hide Aikawa[1], Rina Okada[2], Joshua N. Johnstone [4], Taro Onuma[1,2], Matthew D. W. Piper [4] & Masayuki Miura [1] ✉

Methionine restriction (MetR) extends lifespan in various organisms, but its mechanistic understanding remains incomplete. Whether MetR during a specific period of adulthood increases lifespan is not known. In *Drosophila*, MetR is reported to extend lifespan only when amino acid levels are low. Here, by using an exome-matched holidic medium, we show that decreasing Met levels to 10% extends *Drosophila* lifespan with or without decreasing total amino acid levels. MetR during the first four weeks of adult life only robustly extends lifespan. MetR in young flies induces the expression of many longevity-related genes, including *Methionine sulfoxide reductase A* (*MsrA*), which reduces oxidatively-damaged Met. *MsrA* induction is *foxo*-dependent and persists for two weeks after cessation of the MetR diet. Loss of *MsrA* attenuates lifespan extension by early-adulthood MetR. Our study highlights the age-dependency of the organismal response to specific nutrients and suggests that nutrient restriction during a particular period of life is sufficient for healthspan extension.

Lifelong dietary restriction (DR) is a robust intervention for extending lifespan in various model organisms[1–3]. Many epidemiological and biological data suggest that DR can also be beneficial in humans[4,5]. However, this may depend on age. Lower dietary protein intake can reduce overall mortality in those 65 years or younger, but not in those older than 65 years[6]. A large-scale lifespan analysis using eight hundred mice demonstrated that DR only in the period of 3-24 months can extend lifespan, although the mortality rate is acutely increased upon swapping diet-restricted animals to normal chow[7]. In contrast, DR started at 24 months exerts a blunted transcriptome response, especially in the fat tissues, and results in a minor increase in lifespan, suggesting nutritional memory[7].

*Drosophila melanogaster*, due to its genetic accessibility and techniques for dietary manipulation, is a useful tool for ageing

research[8,9]. In *Drosophila*, the effect of DR was reported to be reversible, hence swapping DR flies to a fully-fed diet on day 14 or 22 almost completely reversed their decrease in mortality[10]. However, a high-sugar diet for the first two-to-three weeks in adulthood showed a long-lasting effect on gene expression and lifespan, suggesting a memory of at least some components of the early adult diet in *Drosophila*[11]. Like DR, intermittent fasting (IF) is another dietary regimen for extending longevity. A two-day fed: five-day fasted IF regime only in the first month of adult life is sufficient for extending *Drosophila* lifespan[12]. Likewise, brief rapamycin treatment in the first two weeks can increase female lifespan via prolonged activation of intestinal autophagy[13]. These reports together raise the question of to what extent and by which mechanisms the organism is affected by early-adult dietary condition.

[1]Department of Genetics, Graduate School of Pharmaceutical Sciences, The University of Tokyo, Bunkyo-ku, Tokyo 113-0033, Japan. [2]Laboratory for Nutritional Biology, RIKEN Center for Biosystems Dynamics Research, Kobe, Hyogo 650-0047, Japan. [3]Laboratory of Molecular Cell Biology and Development, Graduate School of Biostudies, Kyoto University, Kyoto 606-8501, Japan. [4]School of Biological Sciences, Monash University, Clayton, VIC 3800, Australia. [5]These authors contributed equally: Hina Kosakamoto, Fumiaki Obata. ✉e-mail: fumiaki.obata@riken.jp; miura@mol.f.u-tokyo.ac.jp

Methionine (Met), an essential amino acid, is a dietary requirement for animals. Specific restriction of dietary Met has been reported to extend lifespan in rodents[14,15]. Subsequent studies demonstrated that Met restriction (MetR) can extend lifespan in many model organisms, including *Drosophila*[16–19]. It has been suggested in *Drosophila* that active Met metabolism is a key mechanism for longevity[20–23]. Although several possible mechanisms by which dietary Met could regulate the healthspan of organisms, how MetR remodels organismal physiology is not fully understood[24,25]. In addition, whether short-term MetR, especially in early adulthood, can influence organismal lifespan has not been fully studied.

Here, we show in *Drosophila* that MetR in early adulthood can extend organismal lifespan while MetR in later life is ineffective to do so. Using tissue-specific and single-cell RNAseq analyses, we delineate how the early-adult MetR impacts the transcriptome and tissue homoeostasis. Lifespan extension is mediated by sustained induction of the MetR-specific response gene *Methionine sulfoxide reductase A* (*MsrA*). This study reveals a critical time window and mechanism by which specific amino acid restriction achieves lifespan extension.

## Results

### Methionine restriction robustly extends fly lifespan

In *Drosophila*, Met restriction (MetR) extends female lifespan, especially when total amino acid (AA) levels are reduced to 40% of the control[19]. First, we tested whether MetR in our laboratory condition could reproducibly extend lifespan. In our study, we utilised an exome-matched version of the holidic medium developed by Piper et al.[26] and decreased the concentration of all AA to 40% of the original recipe. For Met restriction (MetR), the Met concentration was further decreased to 10% of the control, which resulted in 0.16 mM Met in the MetR diet and 1.6 mM Met in the control diet. This resulted in a negligible (less than 1%) change in the total energy content of the diet from 95.36 kcal/L for the control, to 94.49 kcal/L for MetR. Under MetR, we observed an increase in the median lifespan of wild-type Canton-S females of up to 34.5% (Fig. 1a). This beneficial effect of the MetR diet was also observed in the outbred strain $w^{Dah}$ (Fig. 1b), showing the robustness of the phenotype.

Next, we quantified fecundity (egg-laying) and resistance to stressors. Flies fed with the MetR diet for one week showed a decrease in the number of eggs and a marked increase in starvation resistance in both Canton-S and $w^{Dah}$ (Fig. 1c–f). This set of phenotypes is reminiscent of what has been observed in flies under dietary restriction (DR)[27]. It has been reported that DR increases starvation resistance via increased lipids in the gut[28]. As expected, we observed accumulation of lipid staining in the gut during MetR (Supplementary Fig. 1a, b). MetR also enhanced resistance to an oxidant, hydrogen peroxide, at least in Canton-S (Fig. 1g). Increased stress resistance and decreased fecundity are also frequently reported in long-lived mutants with reduced activity of the insulin/IGF-1 signalling (IIS) pathway[29–31]. Therefore, our MetR condition recapitulates typical long-lived phenotypes.

Recently, it was reported that dietary cholesterol limitation is a strong modifier of lifespan extension by DR[32]. Since our holidic medium in the initial experiments contained 0.1 g/L cholesterol, we increased its concentration to 0.3 g/L (the level found to be adequate in the study[32]). The lifespan extension and increased starvation resistance imparted by MetR were still observed, supporting the conclusion that the present dietary regimen robustly increased fly lifespan and starvation resistance independently of the cholesterol concentration (Supplementary Fig. 1c,d).

Another characteristic of *Drosophila* ageing is a loss of climbing ability[33]. Measuring climbing ability is utilised as a way of measuring healthspan, and both DR and reduced IIS can improve ageing related climbing ability[34,35]. To quantify the climbing ability of individual flies, we developed a negative geotaxis assay (see Methods). Canton-S flies fed with the MetR diet for four weeks showed significantly improved climbing ability, although this phenotype was not observed in $w^{Dah}$ female flies (Supplementary Fig. 1e). Taken together, our data suggest that the restriction of dietary Met recapitulates the benefits of protein restriction and thereby enhances the flies' healthy lifespan.

### Methionine restriction decreases Met metabolites

Met is a precursor of S-adenosylmethionine (SAM), a versatile methyl donor required for various methyltransferases (Fig. 1h). During methylation reactions, S-adenosylhomocysteine (SAH) is produced from SAM. SAH is further metabolised into homocysteine and then enters the transsulfuration pathway to make cysteine via cystathionine. On the other hand, free Met can be converted into an oxidatively damaged form (methionine sulfoxide, MetSO). From LC–MS/MS analysis of these metabolites, we confirmed that whole-body Met levels were decreased under one week of MetR (Fig. 1i). Further, all detected Met metabolites were decreased (Fig. 1j–m). The homocysteine level was below the detection limit in our analysis. Interestingly, the extent to which each metabolite decreased upon MetR largely varied. For instance, SAM and SAH levels were only decreased to half of the control level, while the Met level was reduced to 6.34% of the control (Fig. 1i–k). In contrast, the level of MetSO was sharply downregulated to 0.227% of the control upon MetR. This represents a 28-fold decrease in MetSO when compared to the decrease in internal Met, suggesting the presence of an active mechanism to downregulate MetSO levels during MetR. Considering that the absolute level of MetSO in control animals was ten times greater than that of Met, a large amount of Met was physiologically damaged in the control condition compared to MetR (Fig. 1m).

We also noticed that other AA levels were affected during MetR. Threonine, asparagine, glutamine, and glycine were increased, and leucine, phenylalanine, tryptophan, and tyrosine were decreased (Supplementary Fig. 1f). Although we do not know the mechanisms by which other AAs are increased or decreased by MetR, methionine metabolism is coupled to these amino acids via one-carbon metabolism and mitochondrial metabolism[36–38].

### Early-life MetR extends lifespan

Next, we asked whether MetR in early or late adult life influences lifespan. For this experiment, the flies were fed with the MetR diet only for four weeks, and with the control holidic medium for the rest of their lives. Interestingly, feeding the MetR diet only for four weeks (day5-32) extended lifespan, almost as effectively as lifelong MetR, in female Canton-S (Fig. 2a). In contrast, flies fed with the control diet in the first four weeks did not live longer than non-restricted controls even though Met was restricted for the rest of their lives (Fig. 2a). This phenomenon was also reproduced in the $w^{iso31}$ fly strain using a slightly different MetR condition (1 mM Met in control vs 0.15 mM Met in MetR) (Fig. 2b, c), which was used in a previous report[19]. In this experiment, we restricted Met for the same duration (day5–day32 for early MetR and day32–day58 for late MetR) to compare its effects in young and old flies. Analysing the data using cox proportional hazards revealed that the timing of methionine restriction changed its effects on lifespan; early MetR extended lifespan relative to non-restricted controls, while MetR later in life did not (Fig. 2b, c; Supplementary Table 1). Interestingly, when we used the same protocol on $w^{Dah}$ females, both early and late MetR had similar effects in that they both extended lifespan, although early MetR again had a stronger effect on median lifespan (14.5% increase) than later life MetR (9.7% increase) (Fig. 2d, e; Supplementary Table 1). Importantly, upon returning MetR flies to the non-restricted control diet, the internal Met level rapidly recovered back to the control level in $w^{iso31}$, negating the possibility that the Met level is irreversibly decreased by early MetR (Fig. 2f). Therefore, MetR in early life can robustly increase the lifespan of females with various genetic backgrounds and this effect is diminished or lost when MetR is applied later in life.

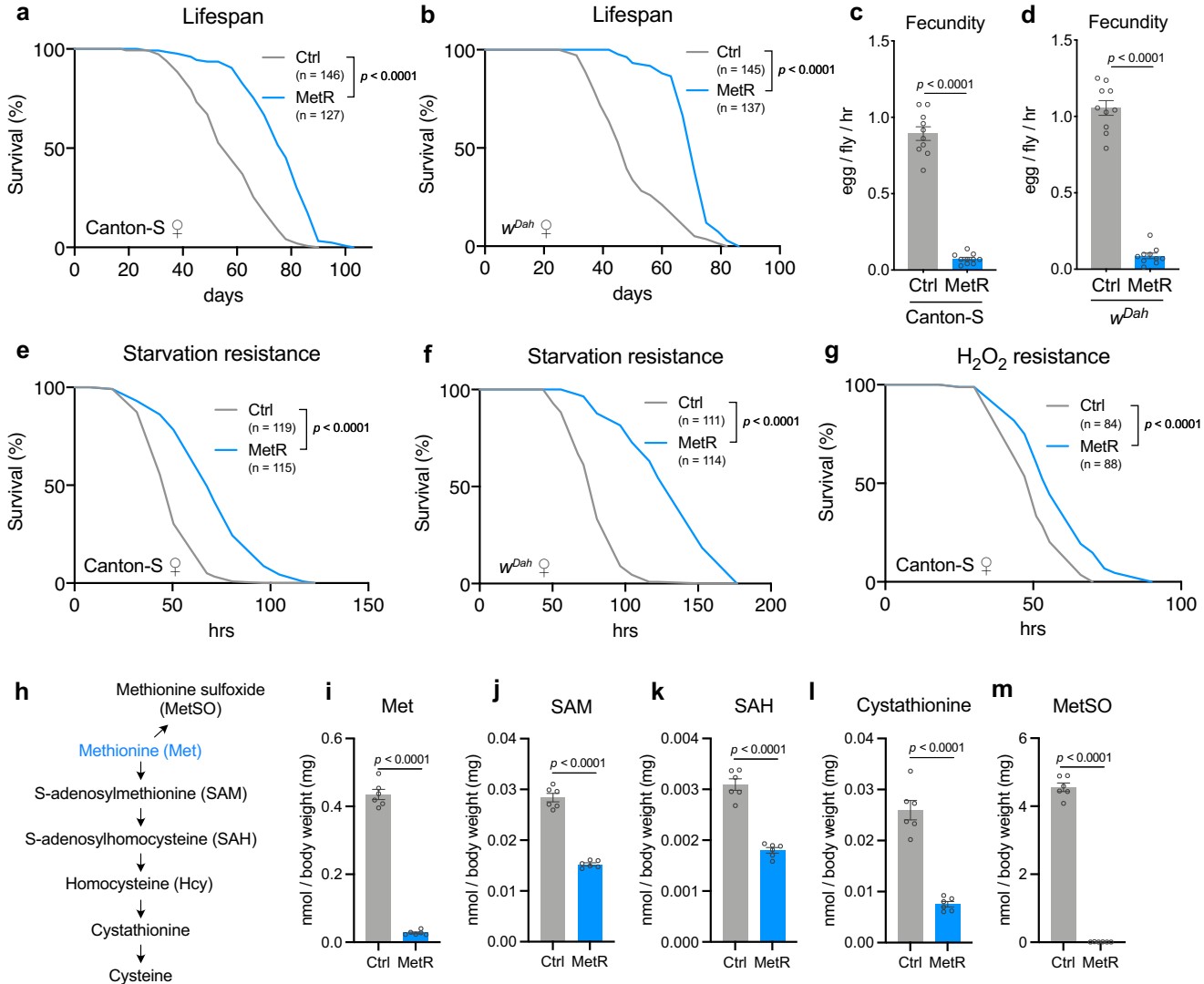

**Fig. 1 | Methionine restriction decreases its metabolites and extends female lifespan. a, b** Lifespans of female Canton-S (**a**) and $w^{Dah}$ (**b**) flies fed with or without a methionine-restricted diet. Sample sizes (n) are shown in the figure. For the statistics, a log-rank test was used. **c, d** Fecundity of Canton-S (**c**) and $w^{Dah}$ (**d**) flies fed with or without a methionine-restricted diet. $n = 10$. For the statistics, a two-tailed Student's $t$ test was used. **e, f** Survivability of female Canton-S (**e**) or $w^{Dah}$ (**f**) flies upon complete starvation after feeding with or without a methionine-restricted diet for one week. Sample sizes (n) are shown in the figure. **g** $H_2O_2$ resistance of female Canton-S flies after feeding with or without a methionine-restricted diet for one week. Sample sizes (n) are shown in the figure. **h** Methionine metabolic pathway and its oxidation to methionine sulfoxide. **i–m** Quantification of methionine metabolites and the oxidative product upon methionine restriction for one week. $n = 6$. For the statistics, a two-tailed Student's $t$ test was used. For all graphs, the mean and SEM are shown. Data points indicate biological replicates. Source data are provided as a Source Data file.

In most cases, male flies do not exhibit large lifespan responses to dietary restriction. We did not observe any lifespan extension in response to early or late MetR in $w^{iso31}$ males (Fig. 2g, h). We also tested the lifespan in $ovo^{D1/+}$ mutant female flies, in which no egg production was observed. The mutant had a relatively longer lifespan than fertile females in the control diet, and MetR in the first four weeks did not increase female lifespan further, suggesting that reproduction is indispensable for lifespan extension during early MetR (Fig. 2i).

A recent study has shown that overexpression of bacteria-derived methioninase in *Drosophila* can break down internal Met and increase lifespan without decreasing the level of other amino acids[39]. To assess if we observed extended lifespan because of reduced methionine or via an interaction between reduced methionine and other amino acids that we modified, we analysed lifespan of female Canton-S flies fed with holidic media containing 100% AA (containing 4 mM Met) with either 10% (0.4 mM Met) or 4% (0.16 mM Met, which is also equivalent to 10% Met when 40% AA is used) of Met levels throughout life or only

in early life. We found that both early and lifelong restriction of Met to 10% in 100% AA background can extend lifespan, but lifelong restriction of Met to 4% cannot (Fig. 2j). Intriguingly, limiting 4% Met restriction to early life only fully extended lifespan (Fig. 2j). This striking observation suggested that 1) decreasing all AA to 40% is not mandatory for MetR-longevity and 2) harsher MetR can extend lifespan, but only when limited to early adulthood. Taken together, these data suggest that there might be a critical window for MetR to exert its maximal benefit and that early MetR likely induces a prolonged effect on physiology.

**Transcriptomic response to dietary methionine declines in aged flies**

Given that the later MetR has only a minor impact on fly lifespan, we speculated that aged flies become less responsive to the MetR diet. To determine how young and old flies react to dietary Met, we performed an RNAseq analysis using the female gut. We targeted the gut for this

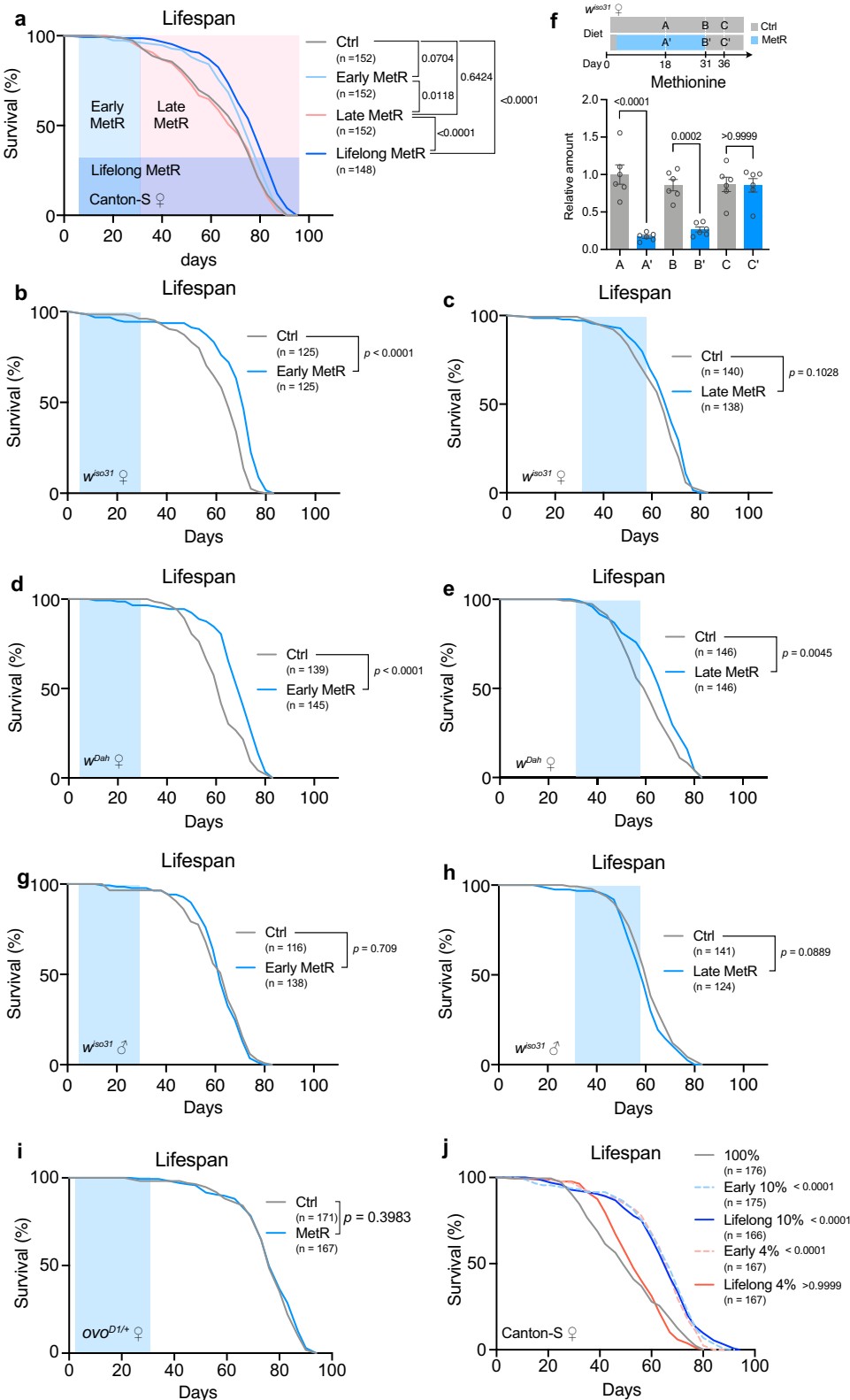

**Fig. 2 | Early adult methionine restriction extends female lifespan. a** Lifespans of female Canton-S flies fed with or without a methionine-restricted diet during their early life (Day 5-32), late life (Day 32-), or whole life (Day 5-). Sample sizes (n) are shown in the figure. For the statistics, a log-rank test was used. **b**–**e**, Lifespans of female flies of $w^{iso31}$ (**b, c**) or $w^{Dah}$ (**d, e**) fed with or without a methionine-restricted diet in early (Day 5-32) life (**b, d**) or late (Day 32-58) life (**c, e**). Sample sizes (n) are shown in the figure. For the statistics, a log-rank test was used. **f** Quantification of methionine upon its restriction over several time courses. n = 6. For the statistics, one-way ANOVA with Holm-Šídák's multiple comparison test was used.

**g**, **h** Lifespans of male flies of $w^{iso31}$ fed with or without a methionine-restricted diet in early life (**g**) or late life (**h**). Sample sizes (n) are shown in the figure. For the statistics, a log-rank test was used. **i** Lifespans of female $ovo^{D1/+}$ flies fed with or without a methionine-restricted diet. Sample sizes (n) are shown in the figure. For the statistics, a log-rank test was used. **j** Lifespans of female Canton-S flies fed a methionine-restricted (10% or 4%) diet compared to 100% AA during their early life or whole life. Sample sizes (n) are shown in the figure. For the statistics, a log-rank test was used. For the graph, the mean and SEM are shown. Data points indicate biological replicates. Source data are provided as a Source Data file.

analysis because previous studies have demonstrated it to be critical for lifespan responses to DR[28,40]. To maximise the response, we completely removed Met (Met-) from the diet and compared it with the complete holidic medium, which included the original amounts of all amino acids. Principal component analysis (PCA), quantification of differentially expressed genes (DEGs, FDR < 0.01, fold change > 1.5), and heatmap analysis clearly demonstrated that the Met- diet induced a strong transcriptional shift in the young gut that was much blunted in the aged gut (Fig. 3a–c, Supplementary Data 1). In the PCA, the PC1 axis separates the animals by age, which was slightly shifted towards the left by just one day of feeding of the Met- diet in young flies, but not in aged flies (Fig. 3a). This suggested that the Met- diet can acutely shift the transcriptome towards a more youthful physiological age. It is also clear that dietary treatment clearly separates the samples along the secondary PC axis in young flies, but this separation is almost eliminated in aged flies.

We found 820 DEGs upon 24-hour feeding of the Met- diet in the young female gut. In clear contrast, we found only 49 DEGs in the guts of aged females fed the same Met- diet, indicating a 94% reduction in the number of DEGs during ageing (Fig. 3b). Gene Ontology analysis revealed that the short-term Met depletion changed gene sets related to "determination of adult lifespan". Several genes included in the GO term were induced by early, but not late, Met depletion (Figs. 3d–f, Supplementary Fig. 2a). These data suggest that the aged gut was much less responsive to the lack of dietary Met. The upregulated genes included *Thor/4E-BP*, a negative regulator of energy-consuming protein synthesis, which is a common target of many nutrient-responsive pathways. Increased *Thor* is a hallmark of the starvation response, suggesting that the tissue sensed the lack of nutrients in the Met- diet, but the older gut did not. Other starvation-related processes, such as autophagy, stress response, lipid metabolism, nutrient transport, and mTOR inactivation were also enhanced in young but not aged guts (Supplementary Fig. 2b–k).

One simple possible reason for the blunted response to MetR in the aged flies is that their internal Met levels might not have been decreased to the same extent as in young flies. To test this hypothesis, we measured Met and its metabolites by LC-MS/MS analysis (Fig. 3g–r). The whole-body Met levels in both young and aged flies were decreased by 24-hour feeding with the Met- diet (Fig. 3g, m). We noticed that many Met metabolites, such as SAM, SAH, and MetSO, were also decreased by the Met- diet in young flies (Fig. 3h, i, k, l), but the effects were mild in the aged flies (Fig. 3n, o, q, r). Nonetheless, the level of cystathionine was similarly decreased even in the aged flies (Fig. 3j, p). The reason why cystathionine can be decreased without changing SAM and SAH levels is unknown, but one can assume that another branch of Met metabolism, such as the Met salvage pathway, in which SAM is converted into polyamines, could be inhibited in the aged flies. In any case, this observation negates the possibility that all Met metabolism was simply slowed down during ageing. Taken together, these results indicate that in aged flies, MetR decreases the levels of internal Met and some Met metabolites but provokes only a partial transcriptomic response.

## Transcriptome analysis of the gut and the fat body to the MetR diet

Complete depletion of dietary Met resulted in a large shift in the transcriptome (Fig. 3a–c). To understand the transcriptional response to early MetR, we performed a time-series 3'mRNAseq analysis in the gut at six hours, two weeks, and four weeks of the lifespan-extending MetR diet. Female *w^Dah* flies fed a 0.15 mM Met diet were compared with age-matched control flies fed with 1 mM Met. The results showed that approximately 30 genes were differentially expressed under MetR conditions at six hours or two weeks (Fig. 4a, Supplementary Data 2). In contrast, four weeks of MetR increased the number of DEGs to as many as 252 (Fig. 4a). There was

no gene commonly changed at all three timepoints. Eighteen upregulated genes were shared by two week- and four-week MetR, while one (six hours vs four weeks) or two (two weeks vs four weeks) downregulated genes were common (Fig. 4b–f). Amongst these genes in common to more than one timepoint, the upregulated genes included *methionine sulfoxide reductase A* (*MsrA*, also known as *Eip71CD*), transporters (*CG3036* and *CG8785*), genes related to lipid metabolic processes (*Lipid storage droplet-2* (*Lsd-2*) and *Acetyl Coenzyme A synthase* (*AcCoAs*)), and serine proteases (*CG18180* and *Jonah 99Fii* (*Jon99Fii*)) (Fig. 4d). Both *MsrA* and *Lsd-2* were also upregulated under the Met- conditions (Figs. 3f, Supplementary Fig. 2e), suggesting that these genes are robustly responsive to dietary Met. The downregulated genes were *Glutathione S transferase E7* (*GstE7*), *CCHamide-2* (*CCHa2*), and an unknown gene *CG3902* (Fig. 4e, f). CCHa2 is an appetite-regulating peptide that is known to promote systemic insulin signalling[41]. Thus, decreased expression of *CCHa2* may lead to a pro-longevity state triggered by reduced IIS activity. Interestingly, we found from i-*cis*Target[42] that many genes containing the foxo consensus binding site were induced in the gut upon four weeks of MetR, further indicating relevance to IIS (Fig. 4g).

To elucidate the systemic response to the MetR diet, we also performed RNAseq analysis of the abdomen upon four-week MetR, which contains the fat body, the major metabolic organ in *Drosophila*. The digestive tract and reproductive organs were carefully removed for this analysis. Strikingly, the most upregulated gene in the abdomen was *MsrA* (Fig. 4h) as was the case in the gut (Fig. 4d). We also noticed that a lipase *brummer* (*bmm*) was upregulated in the tissue, which implied altered lipid metabolism in the flies fed the MetR diet for four weeks. *bmm* is another foxo target, suggesting that foxo is activated in both the gut and the abdomen[43,44].

In contrast, *glycine N-methyltransferase* (*Gnmt*) was strongly suppressed in the abdomen upon MetR (Fig. 4i). Gnmt is an enzyme that produces sarcosine by utilising SAM, and thereby regulates SAM levels[45]. The phenomenon that *Gnmt* expression is suppressed by low methionine intake was observed consistently in our previous study[21]. This SAM buffering system contributes to maintaining SAM levels by avoiding excess SAM utilisation during MetR (Figs. 1j, 3h). Considering that inhibiting *SAM synthase* (*Sam-S*) also results in a decrease in *Gnmt* expression[46], there must be a system to downregulate *Gnmt* mRNA upon detection of decreased SAM levels. The mechanism of this SAM-dependent regulation of *Gnmt* remains to be elucidated.

## MetR induces MsrA in a foxo-dependent manner

Given that *MsrA* was the most upregulated gene by four weeks of MetR in both the abdomen and the gut, we focused on its regulatory mechanism and function. Msr is an evolutionarily conserved antioxidant enzyme required for repairing oxidatively damaged Met (MetSO) back into functional Met[47,48]. It reduces both free and protein-based Met-*S*-SO. In usual physiological conditions, free-MetSO exists in haemolymph at a level four times more than Met (Supplementary Fig. 3a). This implies that Met is oxidised in body fluids, or MetSO formed in tissues is circulated throughout the body. It has already been shown that overexpression of *MsrA* can confer oxidative stress resistance and extend lifespan in *Drosophila*[49]. Ectopic expression of a yeast *Msr*, which can reduce free Met-*R*-SO, extends *Drosophila* lifespan, especially under a Met-rich diet[50]. From these previous reports, we speculated that induction of *MsrA* is one of the functional genes mediating the healthspan-extending effect of MetR.

We first confirmed by quantitative RT−PCR that the induction of *MsrA* was robustly observed when using a lifespan-extending MetR regimen (i.e., 1.6 mM for Control vs. 0.16 mM for MetR in *w^Dah*, Fig. 5a–c). *MsrA* could be upregulated as early as three days of MetR in both the gut and abdomen of males and females (Supplementary Fig. 3b–e). Given MetR-longevity is marginal in males, the MetSO

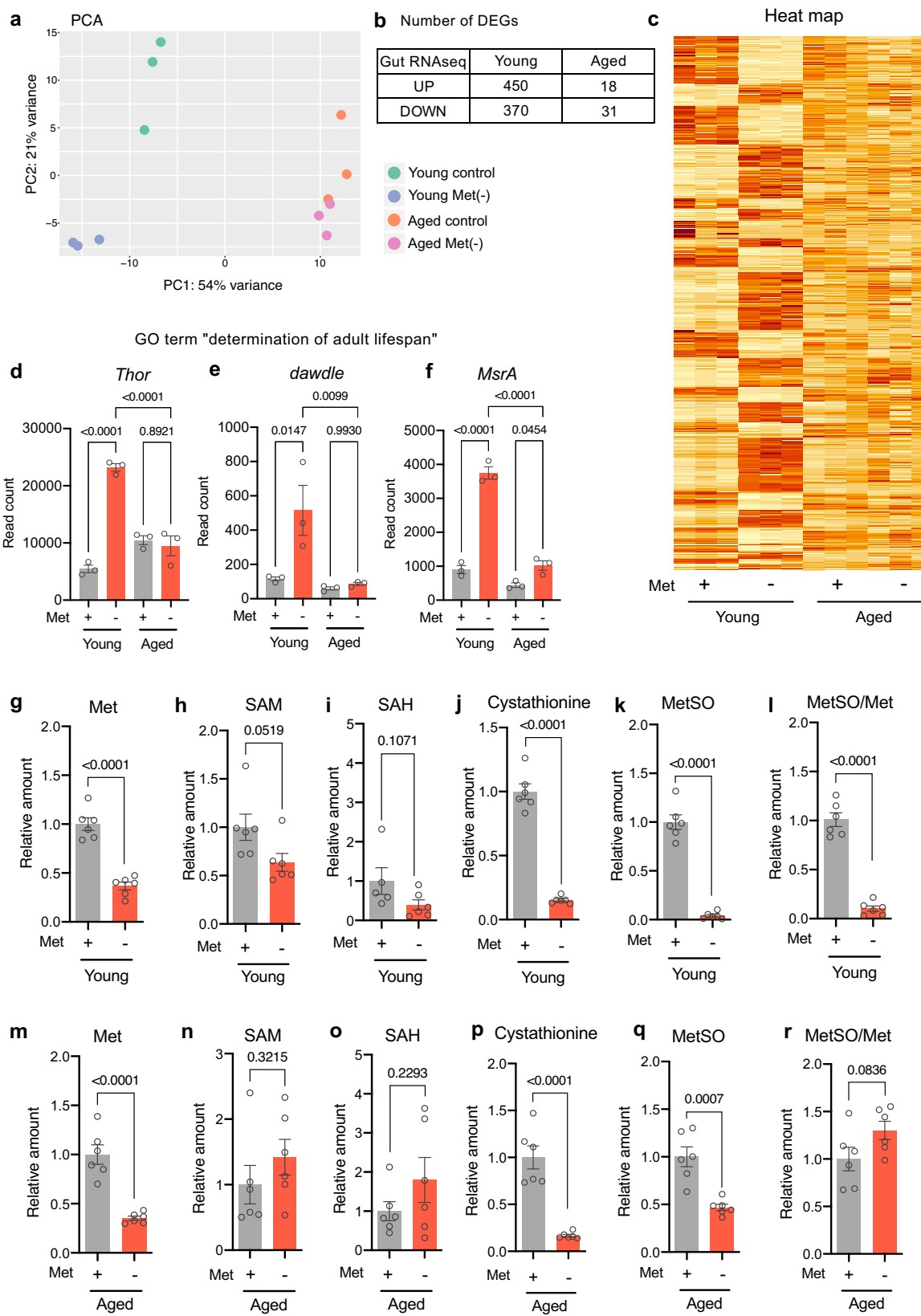

**Fig. 3 | A blunt transcriptomic response to short-term methionine depletion in old flies. a–c** Principal component analysis (**a**), the number of differentially expressed genes (**b**), or heatmap analysis (**c**) of RNAseq analysis of intestines in young (1-week-old) and aged (8-week-old) female $w^{iso31}$ flies upon methionine depletion for 24 hours. **d–f** Read counts of genes categorised as the Gene Ontology of "determination of adult lifespan" in the RNAseq analysis of intestines in young and aged female $w^{iso31}$ flies upon methionine depletion for 24 h. $n = 3$. For the statistics, one-way ANOVA with Holm-Šídák's multiple comparison test was used. **g–r** Quantification of methionine metabolites and the oxidative product upon methionine depletion for 24 h in young (1-week old) (**g–l**) or aged (7-week old) (**m–r**) female $w^{iso31}$ flies. $n = 6$. For the statistics, a two-tailed Student's $t$ test was used. For all graphs, the mean and SEM are shown. Data points indicate biological replicates. Source data are provided as a Source Data file.

**a**

| Gut RNAseq | 6hr | 2week | 4week |
|---|---|---|---|
| UP | 5 | 22 | 131 |
| DOWN | 28 | 4 | 121 |

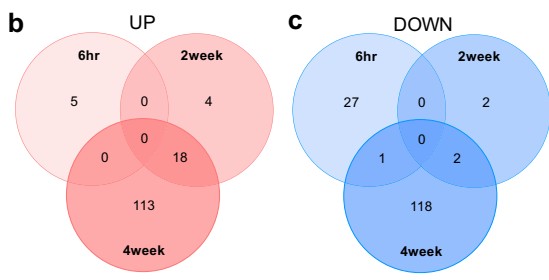

**b** UP

**c** DOWN

**g**

| Possible TF | NES | Logo |
|---|---|---|
| EcR | 6.40733 | transfac_pro__M08954 |
| foxo | 4.64858 | transfac_pro__M07286 |
| CrebB | 4.37878 | jaspar__MA0018.2 |

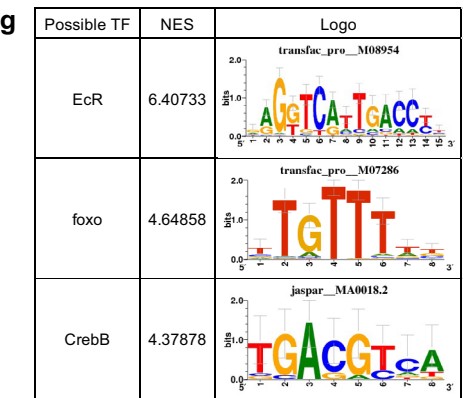

**d** Gut RNAseq

| symbol | name | 2week | | 4week | |
|---|---|---|---|---|---|
| | | log2 FC | padj | log2 FC | padj |
| MsrA | Methionine sulfoxide reductase A | 2.54 | 1.73.E-02 | 2.58 | 4.06.E-03 |
| asRNA:CR45604 | antisense RNA:CR45604 | 1.52 | 6.60.E-03 | 1.87 | 7.55.E-06 |
| CG15784 | uncharacterized protein | 1.28 | 2.89.E-02 | 1.16 | 2.12.E-02 |
| CG3036 | uncharacterized protein | 0.82 | 1.82.E-02 | 1.00 | 2.49.E-04 |
| Lsd-2 | Lipid storage droplet-2 | 0.65 | 1.42.E-02 | 0.97 | 8.12.E-07 |
| CG8785 | uncharacterized protein | 0.79 | 1.32.E-02 | 0.97 | 4.41.E-05 |
| CG16721 | uncharacterized protein | 0.99 | 4.46.E-02 | 0.86 | 4.15.E-02 |
| mlt | mulet | 0.64 | 1.42.E-02 | 0.85 | 1.41.E-05 |
| CG18180 | uncharacterized protein | 0.78 | 1.42.E-02 | 0.79 | 2.73.E-03 |
| CG5059 | uncharacterized protein | 0.62 | 1.42.E-02 | 0.76 | 8.23.E-05 |
| CG6770 | uncharacterized protein | 0.66 | 1.45.E-02 | 0.74 | 6.48.E-04 |
| stv | starvin | 0.54 | 1.42.E-02 | 0.71 | 2.62.E-05 |
| bigmax | bigmax | 0.63 | 1.59.E-02 | 0.71 | 8.18.E-04 |
| Glut4EF | Glucose transporter 4 enhancer factor | 0.57 | 3.26.E-02 | 0.70 | 5.13.E-04 |
| Spn | Spinophilin | 0.56 | 9.48.E-03 | 0.70 | 1.91.E-05 |
| AcCoAS | Acetyl Coenzyme A synthase | 0.75 | 5.98.E-03 | 0.56 | 1.58.E-02 |
| Jon 99Fii | Jonah 99Fii | 0.41 | 6.62.E-03 | 0.42 | 5.13.E-04 |
| lqf | liquid facets | 0.37 | 4.17.E-02 | 0.34 | 2.67.E-02 |

**e**

| symbol | name | 6hr | | 4week | |
|---|---|---|---|---|---|
| | | log2 FC | padj | log2 FC | padj |
| GstE7 | Glutathione S transferase E7 | -0.87 | 1.14.E-02 | -0.76 | 9.48.E-03 |

**f**

| symbol | name | 2week | | 4week | |
|---|---|---|---|---|---|
| | | log2 FC | padj | log2 FC | padj |
| CCHa2 | CCHamide-2 | -0.62 | 1.42.E-02 | -1.31 | 6.73.E-13 |
| CG3902 | uncharacterized protein | -0.82 | 1.05.E-02 | -1.30 | 7.61.E-08 |

**h** Abdomen RNAseq

| symbol | name | log2 FC | padj |
|---|---|---|---|
| MsrA | Methionine sulfoxide reductase A | 5.26 | 2.71.E-29 |
| CG15784 | uncharacterized protein | 3.85 | 1.40.E-12 |
| Ugt37A3 | UDP-glycosyltransferase family 37 member A3 | 3.14 | 4.15.E-02 |
| Cyp309a1 | Cyp309a1 | 2.38 | 8.62.E-06 |
| CG3348 | uncharacterized protein | 2.02 | 1.29.E-05 |
| CG11893 | uncharacterized protein | 1.97 | 4.44.E-04 |
| Arc1 | Activity-regulated cytoskeleton associated protein 1 | 1.91 | 2.92.E-08 |
| GstE7 | Glutathione S transferase E7 | 1.81 | 3.27.E-05 |
| AOX1 | Aldehyde oxidase 1 | 1.80 | 3.07.E-08 |
| Est-6 | Esterase 6 | 1.64 | 1.93.E-02 |
| CG5966 | uncharacterized protein | 1.60 | 1.43.E-03 |
| rost | rolling stone | 1.53 | 1.05.E-02 |
| CG16898 | uncharacterized protein | 1.51 | 1.25.E-02 |
| Bin1 | Bicoid interacting protein 1 | 1.00 | 1.35.E-02 |
| bmm | brummer | 0.92 | 4.15.E-02 |
| Nop17l | Nop17 like | 0.91 | 3.74.E-02 |
| GstD1 | Glutathione S transferase D1 | 0.78 | 4.31.E-02 |

**i**

| symbol | name | log2 FC | padj |
|---|---|---|---|
| Gnmt | Glycine N-methyltransferase | -4.17 | 9.89.E-12 |
| CARPB | Carbonic anhydrase-related protein B | -2.41 | 9.13.E-07 |
| CG5493 | uncharacterized protein | -2.00 | 2.55.E-03 |
| CG10621 | uncharacterized protein | -1.75 | 2.44.E-02 |
| lncRNA:CR33942 | long non-coding RNA:CR33942 | -1.73 | 3.23.E-02 |
| RNaseMRP:RNA | Ribonuclease MRP RNA | -1.39 | 4.71.E-02 |
| Galphao | G protein alpha o subunit | -0.98 | 2.87.E-03 |

**Fig. 4 | Time-course transcriptomic responses to methionine restriction.**
**a–c** Numbers (**a**) and Venn diagrams (**b**, **c**) of differentially expressed genes of the 3' RNAseq results from the intestines of female flies upon acute (six hours) or chronic (two or four weeks) methionine restriction. **d** List of commonly upregulated genes in the female intestines after two and four weeks of methionine restriction. **e** Commonly downregulated genes in the female intestines after six hours and four weeks of methionine restriction. **f** Commonly downregulated genes in the female intestines after two and four weeks of methionine restriction. **g** Possible transcription factors related to the transcriptional responses upon methionine restriction analysed by i-cisTarget. **h**, **i** Lists of commonly upregulated (**h**) and downregulated (**i**) genes in the female abdomens after four weeks of methionine restriction. For the statistics, Wald test was used with DESeq2 (**d–f**, **h**, **i**). Log2 FC, Log₂ Fold Change, padj, adjusted p-value.

levels may not be the limiting factor for male lifespan. *MsrA* transcription is upregulated also in other tissues such as the brain or the thorax (muscle enriched), but not in the ovary (Supplementary Fig. 3f), suggesting that the *MsrA* induction by MetR is a general phenomenon in many cells, if not all. The $ovo^{D1/+}$ mutant females did not significantly increase *MsrA* expression upon MetR (Supplementary Fig. 3g), suggesting that *MsrA* induction in females depends on reproductive capacity. Interestingly, *MsrA* induction by four-week MetR persisted for at least two weeks after shifting back to the control diet, although the magnitude of induction decreased (Fig. 5b, c). Notably, the induction of *MsrA* was attenuated during ageing (Fig. 3f). In addition, MetSO was massively decreased by the Met- diet in young animals, whereas it was only mildly reduced in aged ones (Fig. 3k, l, q, r). Therefore, the increased expression of *MsrA* gene and the concomitant decrease of MetSO in response to MetR, in both absolute levels and relative to the level of Met, was much greater in the young flies, which may account for the lack of benefit of MetR at the later age.

Interestingly, depletion of Met from the diet for 24 h increased *MsrA* expression in the gut, but such effect was not seen for depletion of any other single AA from the diet (Fig. 5d). This suggests that *MsrA* is specifically regulated by dietary Met. *MsrA* is a known target of the pro-longevity transcription factor Daf-16 in *C. elegans*, foxo in *D. melanogaster*, and FOXO3a in a cultured human cell line[49,51]. Considering that foxo was predicted to be a mediator of the MetR-responsive transcriptional shift (Fig. 4g), we hypothesised that MetR increases *MsrA* through the activation of foxo. Consistent with this, *MsrA* induction by the Met- diet was blocked by *foxo*-RNAi (Fig. 5e). Furthermore, *foxo* overexpression in the adult gut for four days was sufficient for *MsrA* induction (Fig. 5f). These data indicate that foxo induces *MsrA* during MetR.

**MsrA function is required for MetR-induced longevity**
To test whether MsrA is functionally relevant to MetR-induced lifespan extension, we asked if loss of function of *MsrA* alters lifespan. A mutant fly line $MsrA^{EY05753}$ with a P-element insertion in the second exon of the

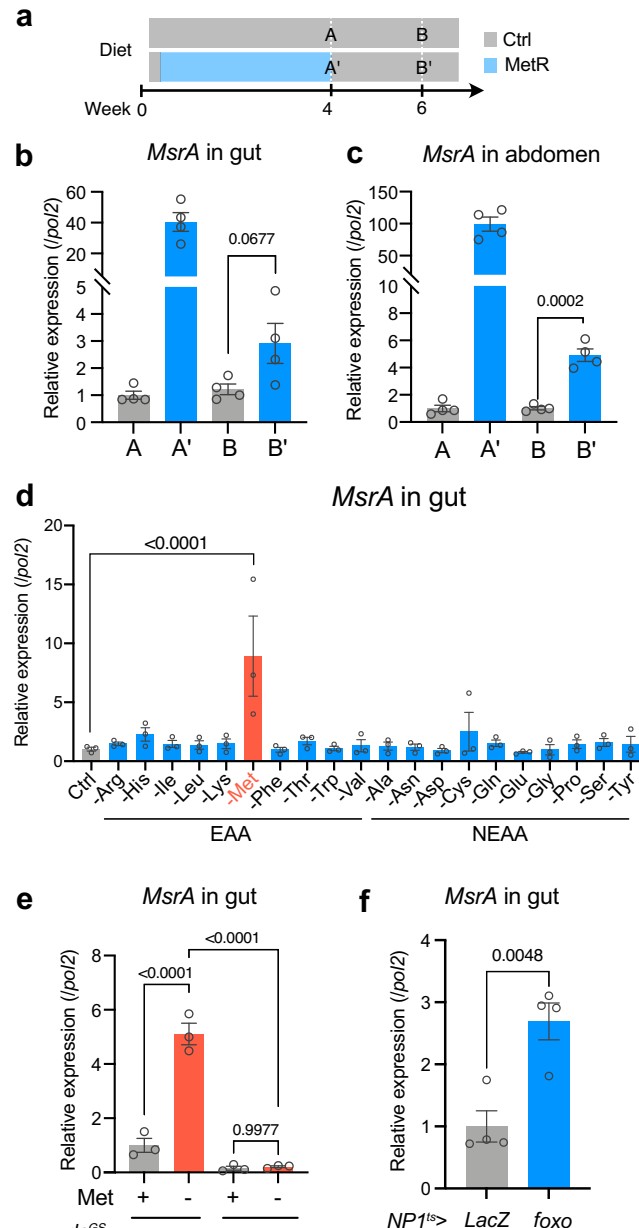

**Fig. 5 | MsrA is induced by methionine restriction via foxo. a–c** Time course (**a**) and the results (**b**, **c**) of the quantitative RT–PCR analysis of *MsrA* in the gut (**b**) and abdomen (**c**). *n* = 4 for (**b**) and (**c**). For the statistics, a two-tailed Student's *t* test was used. **d** Quantitative RT–PCR analysis of *MsrA* in the guts fed with single amino acid-restricted diets for 24 h. *n* = 3. For the statistics, one-way ANOVA with Dunnett's multiple comparison test was used. **e** Quantitative RT–PCR analysis of *MsrA* in the guts fed a methionine-restricted diet for 24 h. Knockdown of *foxo* using the drug-inducible whole-body driver *da^GS*. RU486 (20 μM) was added to the medium. *n* = 3. For the statistics, one-way ANOVA with Holm–Šídák's multiple comparison test was used. **f** Quantitative RT–PCR analysis of *MsrA* in the gut. Overexpression of *foxo* for four days using an enterocyte driver *NP1-Gal4* with temperature sensitive *tub-Gal80^ts*. The flies were fed a standard yeast-based diet. *n* = 4. For the statistics, a two-tailed Student's *t* test was used. For all graphs, the mean and SEM are shown. Data points indicate biological replicates. Source data are provided as a Source Data file.

*MsrA* locus was utilised (Fig. 6a). *MsrA^EYO5753* should be a null allele, as it has a large insertion in the coding sequence common to all isoforms. We confirmed by qRT–PCR that there was no detectable level of *MsrA* transcripts in the mutant (Supplementary Fig. 4a). We backcrossed the *MsrA^EYO5753* line into *w^Dah* to minimise the effect of genetic background

on lifespan and metabolic analyses. The mutant was viable and fertile and did not have a shortened lifespan (compared to *w^Dah*) on the control diet (Fig. 6b). This is consistent with a previous report, demonstrating the redundant functions of MsrA and MsrB[52]. Surprisingly, the lifespan of the *MsrA^EYO5753* female fly is longer than that of *w^Dah* on the control diet (Fig. 6b). This could be because there is an increase in mild oxidative stress in the mutant flies that induces a hormetic effect, reinforcing stress resistance and extending lifespan. Indeed, the mutant showed an increased resistance to hydrogen peroxide (Supplementary Fig. 4b). Strikingly, the *MsrA^EYO5753* mutant did not show lifespan extension upon MetR (Fig. 6b, Supplementary Table 2). Note that, in this analysis, *MsrA^EYO5753* mutant flies have an insertion of the mini *white* gene, while the control flies do not. Thus, while the dose of *white* could have affected lifespan, this cannot account for the change in response to MetR since both red and white-eyed flies have extended lifespan in response to MetR (Fig. 2a, b, d).

In *MsrA^EYO5753* mutants fed with the control diet, we found that Met/SAM/SAH were decreased relative to their levels in wild type controls, while MetSO was increased (Fig. 6c–g). These data suggest that up to half of the free Met is constantly recovered from MetSO under physiological conditions in controls. As observed in Canton-S, all the Met metabolites in *w^Dah* were decreased by one-week-MetR, among which the decrease in the level of MetSO was the strongest (Fig. 1i–m, Fig. 6c–g). This led to a decrease in the MetSO/Met ratio, which was also the case in the Met- diet in young *w^Dah* flies (Figs. 3l, 6g). As anticipated, while Met, SAM, and SAH were still decreased, the MetSO reduction was blunted and the MetSO/Met ratio was increased in the *MsrA^EYO5753* mutant during MetR (Fig. 6g). These data are similar to what we observed when we subjected aged flies to Met depletion (Fig. 3r). Together, these data indicate that the MetR-induced decrease of MetSO requires MsrA in a manner that is consistent with it having a causal role in lifespan extension.

## Lipid metabolism may not be involved in lifespan extension by MetR

MetR promoted lipid accumulation in the gut of wild-type flies (Supplementary Fig. 1a, b) and upregulated the gene expression of *Lsd-2* and *bmm* (Fig. 4d, h, Supplementary Fig. 2e). This phenotype implies that MetR promotes lipid turnover, phenocopying what happened during conventional DR[28,53]. To test whether altered lipid metabolism contributes to early MetR-longevity, we analysed a *bmm^1* mutant and its control animals with the same genetic background[43,54]. In this mutant, neutral lipid in the gut was already abundant under the control diet, and was mildly upregulated upon MetR, which correlated well with the flies' enhanced starvation resistance (Supplementary Fig. 5a,b). Mutation of the *bmm* lipase did not compromise the lifespan extension imparted by MetR (Supplementary Fig. 5c, Supplementary Table 3). Although we cannot rule out the possibility that blocking other lipid metabolic genes or upstream transcription factors could alter MetR longevity, these data suggested that altered lipid metabolism contributed little to the lifespan regulation in this context.

## Blocking the transsulfuration pathway did not abrogate MetR-induced longevity

Methionine metabolism is coupled with cysteine metabolism through the transsulfuration pathway (TSP) (Supplementary Fig. 6a). To test whether TSP is related to MetR-longevity, we used the TSP inhibitor propargylglycine (PPG). First, we fed female flies with various concentrations of PPG during MetR and quantified the internal Met metabolites. As expected, cystathionine accumulated in response to PPG in a dose dependent manner, while the levels of Met, SAM, SAH and MetSO did not respond to the PPG concentration (Supplementary Fig. 6b–f). The administration of PPG shortened female lifespan (Supplementary Fig. 6g). Strikingly, however, we still observed lifespan extension in response to early-adult MetR (Supplementary Fig. 6g).

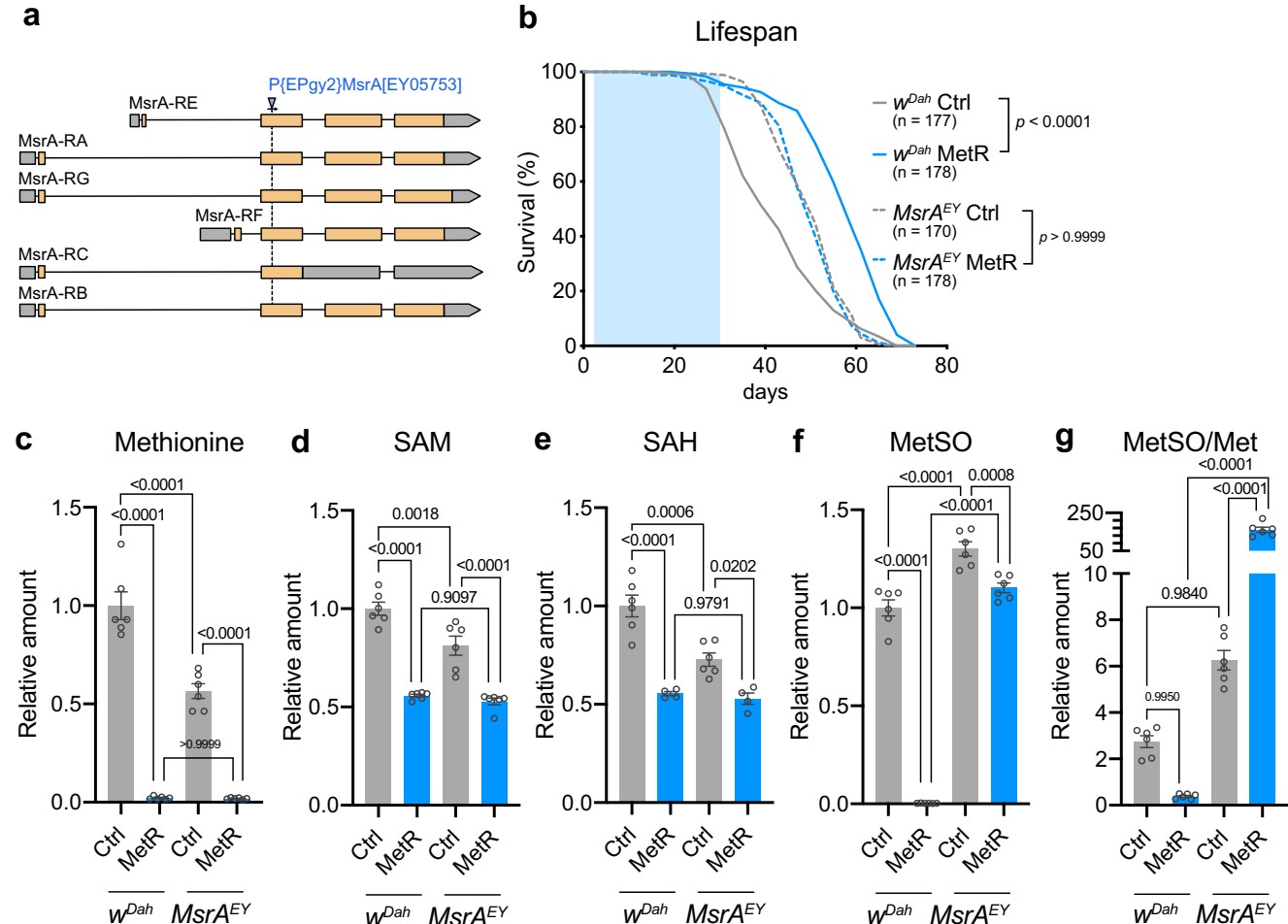

**Fig. 6 | Loss of *MsrA* abolishes lifespan extension by early adult methionine restriction. a** The gene structure of *MsrA* and the insertion site of the p-element in the *MsrA^EY05753* mutant. **b** Lifespans of female flies of *w^Dah* and *MsrA^EY05753* fed with or without a methionine-restricted diet in early life. Sample sizes (n) are shown in the figure. For the statistics, a log-rank test was used. **c–g** Quantification of methionine metabolites and the oxidative product in female *w^Dah* or *MsrA^EY05753* flies upon methionine restriction. n = 6. For the statistics, one-way ANOVA with Holm-Šídák's multiple comparison test was used. For all graphs, the mean and SEM are shown. Data points indicate biological replicates. Source data are provided as a Source Data file.

From these data, we concluded that TSP is not involved in the MetR-induced longevity.

## Single-cell RNAseq analysis of the gut upon methionine restriction

Lastly, we tried to describe how MetR influences tissue homoeostasis and thereby preserves lifespan by single-cell RNAseq analysis of the female gut. The midgut samples were dissected from *w^Dah* flies in young (day6), aged (day40, 1.6 mM Met for five weeks), and aged-MetR (day40, 0.16 mM Met for five weeks) conditions (Fig. 7a). The Malpighian tubules, the hindgut, and the proventriculus were removed manually. The estimated numbers of sequenced cells for young, aged, and aged-MetR were 10364, 15604, and 20313, respectively.

After batch correction and data cleaning, the cells were clustered into 24 different cell types expressing specific genes, which were visualised using a uniform manifold approximation and projection (UMAP) plot (Fig. 7b, Supplementary Fig. 7a–c). As commonly observed in the fly gut, we found four major cell types: intestinal stem cell (ISC), enteroblast (EB), enterocyte (EC), and enteroendocrine cell (EE). The number of cell clusters in each cell type varies among studies[55–57]. In our case, we observed seven progenitor (ISC/EB) clusters, nine EC clusters, and seven EE clusters. During ageing, the proportions of cell numbers in some clusters changed. For example, there were decreases in several progenitors (EB/ISC, EB1, and EB2), and

increases in some enterocyte clusters (Fig. 7c–e). Among these changes, MetR restored the increased proportions of clusters of posterior enterocytes (pEC1-pEC4) to more youthful levels (Fig. 7d). These results suggest that one of the benefits of MetR can be the suppression of the (relative) expansion of posterior enterocytes.

Next, we investigated what kind of transcriptomic change occurs in each cell cluster (Supplementary Data 3). A heatmap analysis of the DEGs in each cell type showed a trend in which age-related shifts in the transcriptional signature were recovered back by MetR, especially in progenitors and enterocytes (Supplementary Fig. 8a, b). In contrast, MetR induced a distinctive transcriptome pattern in enteroendocrine cells (Supplementary Fig. 8c). PCA of the data also supported this observation (Supplementary Fig. 9). For instance, in progenitors, we observed a shift to the left in the PC1 axis by ageing, which was shifted back to the right by MetR (Supplementary Fig. 9a). This was also the case in anterior and posterior enterocytes, where the age-dependent shift along the PC1 axis was recovered under MetR condition (Supplementary Fig. 9b, d). However, the degree of the rescue of the age-related transcriptomic shifts varied greatly among clusters.

It has been observed that the number of damaged/active ISCs rises in the aged gut[57]. However, in our analysis, we did not observe this rise, but rather decreased numbers of some progenitor cells (Fig. 7c). One possible reason for this is that our flies were fed with the holidic

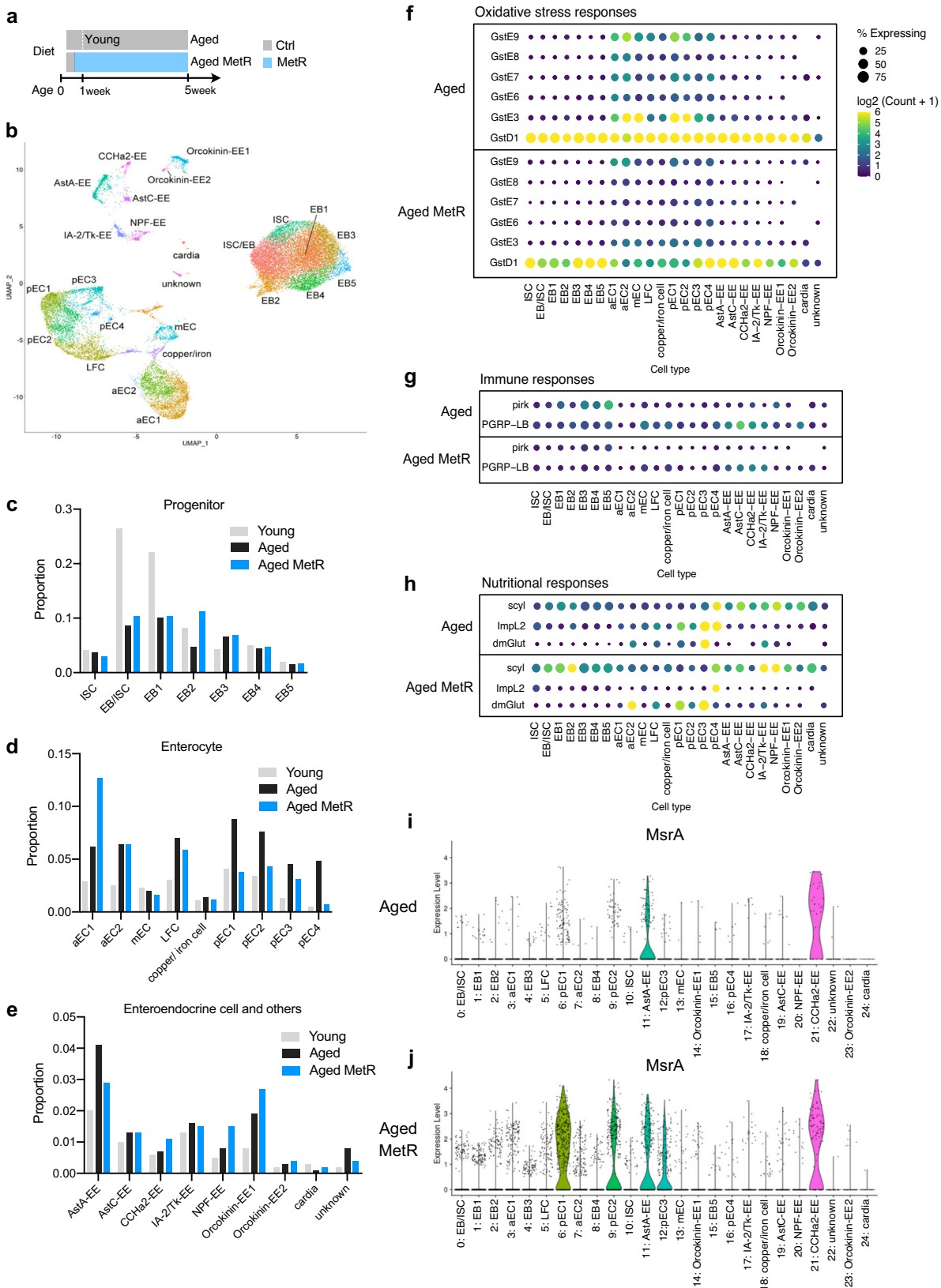

**Fig. 7 | Single-cell RNAseq analysis of the guts of methionine-restricted flies.**
**a** Experimental time course of gut sampling for single-cell RNAseq analysis upon methionine restriction. **b** UMAP plot and annotated cell types from the single-cell RNAseq analysis. **c**–**e** Proportions of cell types in progenitor cells (**c**), enterocytes (**d**), and enteroendocrine cells (**e**). **f**–**h** Dot plots of genes related to immune responses (**f**), oxidative stress responses (**g**), or nutritional responses (**h**). **i**, **j** Violin plots of *MsrA* expression in the aged gut (**i**) or the aged gut with methionine restriction (**j**). Source data are provided as a Source Data file.

medium, which contains more preservatives. This diet generally resulted in lower numbers of gut bacteria, which are known to mediate the age-related increase in ISC expansion[58]. Another possibility is that the data-cleaning process removed atypical cells in the aged samples from the clustering analysis, due to the disturbed gene expression signature. For the Aged sample, only 31.7% of the sequenced cells were clustered (4946/15604), whereas this was 99.5% for the Young sample (10310/10364). For the Aged-MetR sample, this number was increased to 48.6% (9863/20313), perhaps implying that their transcriptional profile was more youthful, and that tissue ageing was delayed.

Considering that there were also large differences in the quality of the data (RNA volume, sequence reads, number of genes) between the Young and the two Aged samples, it seems difficult to fairly compare the absolute gene expression between the two ages. Thus, we carefully looked at DEGs that were changed between Aged and Aged-MetR. We noticed that some DEGs were commonly changed among cell clusters. Several GSTs were decreased by MetR (Fig. 7f), which was consistent with the bulk RNAseq analysis (Fig. 4e). Decreases in GSTs expression imply a lower level of oxidative stress in the tissue. It was also clear that expression of the immune-related genes *pirk* and *PGRP-LB* were decreased by MetR, suggesting attenuation of the inflammatory state in the aged gut (Fig. 7g). Recent scRNAseq analysis in the fly gut also found that suppressing an innate immune Imd pathway activity in ISCs autonomously attenuates age-related ISC overproliferation[59]. We also observed that some nutritional response genes were affected (Fig. 7h). Notably, *MsrA* was induced in many cell types, especially in the posterior enterocyte clusters (Fig. 7i, j). This was correlated with the MetR-driven decrease in the number of cells and GSTs expression in the posterior enterocytes (Fig. 7d, f). Taken together, single-cell RNAseq analysis of the gut suggested that MetR delayed the tissue ageing signature. Our data also provide a resource of many addressable hypotheses to understand how a nutrient impacts the health of each cell type in the tissue.

## Discussion

There is an urgent need to develop anti-ageing interventions. Studies using model organisms have contributed to understanding the biology of ageing, which will eventually lead to the development of various methods of lifespan extension[60]. DR is thus far one of the most robust and practical applications for human society. However, restricting diet throughout life is not entirely appropriate since the effect of diet depends on the conditions, such as age[5]. In this study, we

demonstrated that MetR in early adulthood efficiently extends fly lifespan, whereas that in later adulthood has a milder effect (Fig. 8). This may be at least partly because aged flies cannot trigger the beneficial transcriptional shift in response to the dietary manipulation, despite internal Met levels being decreased, suggesting that ageing blunts the transcriptional response to a particular nutrient shortage. Our data also imply the existence of a "methionine memory", in which the amount of dietary Met to which the animal is exposed in its early life stage determines the degree of lifespan benefit experienced at older ages. They also suggested that MetR can help maintain a younger state, but once the animal's physiological condition goes beyond a threshold or critical period (middle to aged), it may not preserve the animal anymore.

How MetR provokes the (prolonged) transcriptional response and lifespan extension is not entirely understood. Theoretically, decreased Met can be sensed through amino acid-sensing kinases such as general control the General Control Nonderepressible 2 (GCN2) and mechanistic target of rapamycin (mTOR). Activation of mTOR in the fat body by nutrients can stimulate the secretion of Dilps, which in turn activates insulin/IGF-1 like signalling (IIS) systemically[61]. This nutritional relay has been extensively studied in the larval stage, but the mechanism is also thought to be conserved in adults[62–64]. GCN2 activation leads to activation of a transcriptional regulon through activation transcription factor 4 (ATF4), while downregulation of the TOR/IIS pathway activates the transcription factors, REPTOR and foxo[65]. We found that foxo, ecdysone receptor (EcR), and CREB were predicted to be transcription factors involved in the MetR-induced transcriptomic response. Interestingly, both EcR and CREB transcription factors are known to be cofactors of ATF4. Thus, it is likely that these nutrient sensing machineries were all affected by MetR. In our RNAseq analysis of the young gut upon Met depletion, we noticed an upregulation of several genes related to autophagy (Supplementary Fig. 2b), which might be attributable to the foxo activation[35,66]. Given that the lifespan extension by early-life rapamycin treatment in *Drosophila* is reported to be dependent on intestinal autophagy[13] and that MetR in autophagy-defective yeasts has been shown to fail to extend their lifespan[67], upregulation of autophagy in the gut or other tissues could be a key mechanism for MetR-induced longevity programme. It is compelling to explore the mechanisms of the interplay among foxo, MsrA, and autophagy in the context of lifespan extension by early MetR.

In this study, we found that *MsrA* was likely to be regulated by foxo. Increased *MsrA* expression two weeks post MetR suggests

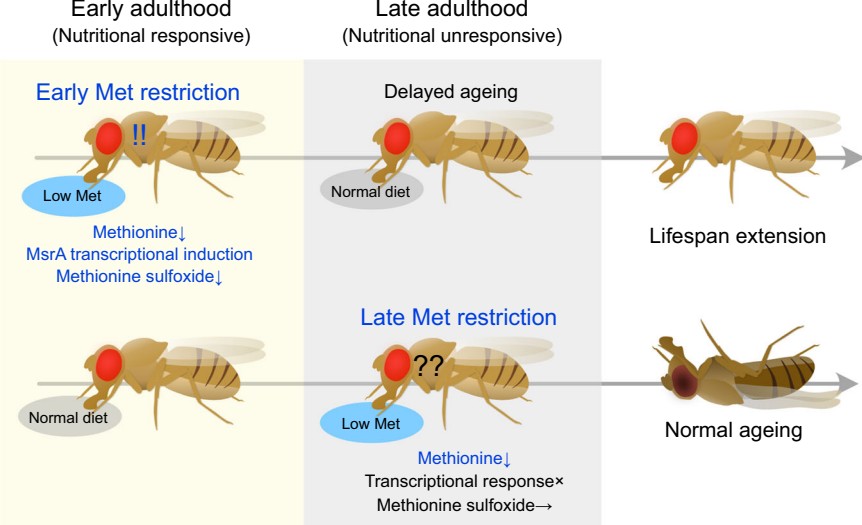

**Fig. 8 | A model of lifespan extension by early Met restriction.** Methionine restriction in young flies can induce *MsrA* and decrease MetSO level, leading to lifespan extension. This nutritional response wanes in older flies, despite internal methionine levels are decreased.

prolonged foxo activation. It is possible that decreased IIS activates foxo under MetR. However, at least in worms, *MsrA* overexpression enhances the nuclear localisation of DAF-16/foxo in the gut. Therefore, a foxo-MsrA feedforward loop can also mediate the effect of MetR. Careful analysis of foxo activation, including protein modification and localisation, is necessary to further understand the mechanistic basis. Additionally, it is intriguing to understand why the foxo-MsrA axis is specifically sensitive to Met, but not other AAs.

There is a general concept that ageing decelerates metabolism. However, Met depletion in both young and aged flies resulted in decreased levels of cystathionine, a downstream metabolite produced through TSP. Nonetheless, we observed increased SAM and SAH levels in aged flies upon Met depletion. Given that SAM seems to be a major regulator of intestinal homoeostasis and organismal lifespan[21,68], a lack of SAM decrease in aged flies would result in a blunted response to Met restriction. SAM is also a precursor for polyamines, which are reported to be mediators of DR-longevity in *Drosophila*[23]. If SAM is a major regulator of the foxo-MsrA axis, we could explain the Met specificity for the regulation of *MsrA* expression. This possibility needs to be carefully analysed in the future.

Our data suggested that *MsrA* is a functional target of MetR that drives the beneficial programme that counteracts organismal ageing. It is reported in the genetic model of MetR that, the level of MetSO is also strongly decreased, although the direct contribution of *MsrA* to this phenotype was not tested[39]. The literature has already demonstrated that MsrA contributes to reducing MetSO and alleviating oxidatively damaged proteins and free non-functional Met. Interestingly, dimethyl sulfide (DMS) is reported to extend lifespan in *C. elegans* and in *Drosophila* in an *MsrA*-dependent manner[69]. In contrast, MsrA was reported to be nonessential for the metabolic benefit of MetR in mice[70]. Whether MsrA is required for lifespan extension in other contexts in various organisms including mammals needs to be tested in the future.

Our study did not identify the tissue where 1) Met is sensed and 2) *MsrA* functions to drive a pro-longevity mechanism. Previous research using genetic MetR models in *Drosophila* has shown that lifespan extension occurs when Met levels are decreased either in the gut or in the fat body[39]. Their data suggest that MetR in at least these two tissues are sufficient for the organismal longevity. Another report indicates that overexpression of *Gnmt* in the fat body has been found to extend lifespan[23]. These results suggest that a decrease in Met/SAM levels in the gut or fat body can be a key factor in lifespan extension. This is consistent with the fact that we found *MsrA* to be prominently induced in these organs during MetR. However, we also found that *MsrA* induction occurred systemically, and that Met/SAM/MetSO are metabolites that can be transported through circulation (as they exist in haemolymph). Thus, changes in *MsrA* expression in multiple peripheral tissues, not only in a specific tissue, may be important to decrease systemic MetSO levels to an extent that extends lifespan.

It is possible that *MsrA* induction during MetR is an adaptation of cells to counteract Met shortage; perhaps MetSO can serve as a store of Met. Indeed, the absolute amount of MetSO is ten times higher than that of free Met in female flies. Interestingly, recent work has shown that egg production in female *Drosophila* decays quickly (within three days) in response to deprivation for any one of eight of the essential amino acids[71]. By contrast, exposure to food lacking either histidine or methionine results in a much slower decay in egg production (seven days) indicating that the flies have internal reserves of methionine that they can call on to buffer against environmental fluctuations. This response may have evolved specifically for these essential amino acids because they appear to be consistently limiting for flies feeding on their natural diet of yeast[18,26,72].

Once we know the mechanism of MetR-induced *MsrA* expression, we would then be able to pinpoint which step of the nutritional response is defective in aged flies. Interestingly, it has been shown that *foxo* remains inside the nucleus even in aged fly tissues, but its target

genes are different from younger fly tissues[44]. Met-dependent changes in chromatin status can be another mechanism, as SAM-dependent modification of histones and nucleic acids should be altered during early MetR[73–75]. Furthermore, it would be intriguing to explore whether the mechanism of this age-dependent transition of the nutritional response is conserved in humans.

## Methods

No ethical approval was obtained because insect models do not require ethical approval under local laws and regulations.

### Drosophila stocks and husbandry

Flies were reared on a standard diet containing 4% cornmeal, 4-6% yeast, 6% glucose, and 0.8% agar with 0.3% propionic acid and 0.15% nipagin. Canton-S, $w^{iso31}$ (from Dr Alex Gould), and $w^{Dah}$ (from Dr Linda Partridge) flies were used as control strains. Adult flies were maintained under conditions of 25 °C and 60-65% humidity with 12 h/12 h light/dark cycles. To allow for synchronised development with constant density, embryos were collected on agar plates (2.3% agar, 1% sucrose, and 0.35% acetic acid) with live yeast paste and directly added to the standard diet in a bottle with a fixed volume. Normally, 150–200 adult flies/bottle were obtained.

The fly lines used in this study were *da-GeneSwitch* (from Dr Monnier Veronique), *NP1-Gal4*, *tub-Gal80^{ts76}*, *UAS-foxo-RNAi* (BDSC, 27656), *bmm^{WT43,54}*, *bmm^{1 43,54}*, *ovo^{D1}* (BDSC, 1309), *UAS-lacZ* (Kyoto, 106500), *UAS-foxo* (Bloomington, 9575), and *MsrA^{EY05753}* (Bloomington, 16671). *MsrA^{EY05753}* were backcrossed eight generations to $w^{Dah}$. Unless otherwise stated, female flies were used for all experiments.

### Fecundity analysis

After flies were raised on the standard yeast diet for two days after eclosion, 15 female and 15 male flies were collectively transferred to the control holidic medium or methionine-restricted medium for mating. Flies were transferred to fresh vials every three days. One week after dietary manipulation, anaesthetised flies were separated into groups of three males with three females. After 24 h, the egg number in each vial was counted.

### Climbing ability

A negative geotaxis assay was used to quantify climbing ability. For each condition, 10–30 flies were transferred to each cuboid vial (w: 2.6 cm d: 1 cm h: 16.4 cm) using a funnel. These vials were placed in front of a light box (HOZAN, 8015-011902) inside a dark tent to shed light from behind. Flies inside were dropped to the bottom of the vial by three consecutive taps. One minute after this practice tap, the actual assay was performed and recorded using a web camera (logicool, 860-000336). The height of the flies ten seconds after the tap was measured manually. Four vials were prepared for each group.

### Dietary manipulations

The exome-matched version of a chemically defined diet, or holidic medium, was used for dietary manipulations[26,77]. Methionine restriction (MetR) is achieved by decreasing the Met concentration. To minimise the batch effect, the medium was prepared without Met and then split into two bottles, followed by the addition of Met stock solution (or just Milli-Q water for the Met(-) diet) to each bottle to make the control and MetR diets. We originally used 1 mM vs. 0.15 mM Met for MetR; however, in the middle of the study, we empirically noticed that 1 mM Met was not adequate for stably producing well-fed physiological conditions. We increased the Met concentration to 1.6 mM, which was 40% of the original concentration of the holidic medium. We confirmed that all results were reproduced well by both MetR conditions. For lifespan analysis in Fig. 2j, we used original (100%) AA concentration which contains 4 mM of Met. All the ingredients used and the procedure to make the holidic medium are

described in the Supplementary Data 4. The food calorie is calculated based on the content of sucrose (3.87 kcal/g), amino acids (4 kcal/g), and agar (0.031 kcal/g).

## Lifespan measurement

Adult flies that eclosed within one day were collected and maintained for an additional two days for maturation and mating on the standard diet. Flies were then sorted by sex and maintained at a fixed density (15 flies/small vial or 30 flies/large vial). For lifespan analysis, the number of dead flies was counted every three to four days when flies were transferred to fresh vials. For each lifespan curve, at least six vials were used in parallel to minimise inter-vial variation. PPG (Sigma, P7888-1G) was added to the diet to inhibit the transsulfuration pathway.

## Starvation/H$_2$O$_2$ resistance analysis

Female flies were transferred to a 1% agar diet (for starvation) or a 3% H$_2$O$_2$, 5% sucrose, 1% agar diet (for H$_2$O$_2$ resistance), and the number of dead flies was counted 2-3 times per day until all flies were dead.

## RNA sequencing analysis and quantitative RT–PCR

For RNAseq analysis the guts of young or old flies upon 24 h Met restriction, total RNA was purified from eight guts. Tissues were homogenised in Qiazol (Qiagen, 79306) using a pellet pestle (bms, BC-PES50S). RNA purification was performed using an RNeasy micro kit (Qiagen, 74004). Triplicate samples were prepared for each experimental group. The RNA samples were sent to Macrogen, where RNA sequencing was performed using an Illumina NovaSeq 6000. The paired-end sequence data were analysed as follows: a quality check of the raw reads was performed by FastQC (v0.11.9). The raw reads were filtered to remove the adaptors and low-quality bases using Trimmomatic (v0.39). Filtered reads were aligned to the *Drosophila* genome (BDGP6.22) using Hisat2 (v2.1.0). The TPM values were calculated using subread (v1.6.5). Differentially expressed genes were identified using DESeq2 (v1.26.0). RNA-sequencing data have been deposited at DDBJ with accession number DRA013585.

For 3′ RNAseq analysis of guts after 6 h, 2 weeks, or 4 weeks of Met restriction, 10 guts of female $w^{Dah}$ flies were dissected. For RNAseq analysis of abdomens upon 4 weeks of Met restriction, 10 abdomens of female $w^{Dah}$ flies were dissected. Triplicate samples were prepared for each experimental group. Met restriction was started on Day 5 after 3 days of feeding with the control holidic medium. Total RNA was purified using the Promega ReliaPrep RNA Tissue Miniprep kit (z6112). RNA was sent to Kazusa Genome Technologies to perform 3′ RNA-seq analysis. The cDNA library was prepared using the QuantSeq 3′ mRNA-Seq Library Prep Kit for Illumina (FWD) (LEXOGEN, 015.384). Sequencing was performed on an Illumina NextSeq 500 using the NextSeq 500/550 High Output Kit v2.5 (75 cycles) (Illumina, 20024906). Raw reads were analysed by the BlueBee Platform (LEXOGEN), which performed trimming, alignment to the *Drosophila* genome (BDGP6), and counting of the reads. The count data were statistically analysed by the Wald test using DESeq2. The results have been deposited in DDBJ under accession number DRA013644. For the prediction of *cis*-regulatory elements, i-*cis*Target was used[42].

For quantitative RT–PCR analysis, total RNA was purified from 4 tissues of female $w^{Dah}$ flies using the Promega ReliaPrep RNA Tissue Miniprep kit (z6112). cDNA was made from 100–400 ng of DNase-treated total RNA using a Takara PrimeScript RT Reagent Kit with gDNA Eraser (Takara bio RR047B). Quantitative PCR was performed using TB Green™ Premix Ex Taq™ (Tli RNaseH Plus) (Takara bio RR820W) and a QuantStudio 6 Flex Real Time PCR system (Thermo Fisher) using *RNA pol2* as an internal control. Primer sequences were: *MsrA*-Forward, gccggttcacgatgtgaatg, *MsrA*-Reverse, gtagcccacggtggttctc, *MsrA$^{EY05753}$* mutation check-Forward, agctcaaggatctgagcacc, *MsrA$^{EY05753}$* mutation check-Reverse, ccgtggctttggtgacattc, *RNA pol2*-Forward, ccttcaggag-tacggctatcatct, and *RNA pol2*-Reverse, ccaggaagacctgagcattaatct.

## Measurement of metabolites

Metabolites were quantified by ultra-performance liquid chromatography–tandem mass spectrometry (LCMS-8060, Shimadzu) based on the Primary Metabolites package ver.2 (Shimadzu)[56,57]. Four whole bodies of female flies were homogenised in 160 µL of 80% methanol containing 10 µM internal standards (methionine sulfone and 2-morpholinoethanesulfonic acid). After centrifugation at 4 °C and 20,000 × $g$ for 5 min, 150 µL of the supernatant was deproteinized by mixing it with 75 µL of acetonitrile. The supernatant was placed into a pre-washed 10 kDa centrifugal device (Pall, OD010C35), and the flow-through after centrifugation at 4 °C and 14,000 × $g$ for 5 min was evaporated completely using a centrifugal concentrator (TOMY, CC-105). Haemolymph was extracted according to the following protocol: A glass capillary was used to prick the side of the female abdomen, after which 10 flies were collected in a 0.5 mL tube that had been pierced at the bottom by a syringe. The resulting sample was then placed on top of a 1.5 mL tube and centrifuged at 4 °C and 8000 × $g$ for 5 min. Next, the collected haemolymph sample in the 1.5 mL tube was mixed and vortexed with 50% methanol containing 10 µM internal standards. Chloroform (250 µL) was added to the sample and vortexed. After centrifugation at 4 °C and 2300 × $g$ for 5 min, 200 µL of the supernatant was deproteinized by mixing it with 100 µL of acetonitrile. The supernatant was placed into a pre-washed 10 kDa centrifugal device (Pall, OD010C35), and the flow-through after centrifugation at 4 °C and 14,000 × $g$ for 5 min was evaporated completely using a centrifugal concentrator (TOMY, CC-105). The dried and concentrated samples were resolubilised in ultrapure water and subjected to LC-MS/MS with a PFPP column (Discovery HS F5 (2.1 mm × 150 mm, 3 µm), Sigma–Aldrich) in a column oven at 40 °C. A gradient from solvent A (0.1% formic acid, water) to solvent B (0.1% formic acid, acetonitrile) was performed for 20 minutes. MRM method parameters were optimised by the injection and analysis of pure standards through peak integration and parameter optimisation with the use of software (Labsolutions, Shimadzu). The concentrations of metabolites were normalised by methionine sulfone and the body weight of flies or protein amount in the haemolymph. For body weight measurement, flies were anaesthetised by CO$_2$ and placed on a microbalance (METTLER TOLEDO, XPR2). The mass spectrometry raw data generated in this study have been deposited in the DDBJ MetaboBank under accession codes MTBKS226, MTBKS227, MTBKS228, MTBKS229, MTBKS230, MTBKS231 [https://ddbj.nig.ac.jp/public/metabobank/study/]. The mass spectrometry data generated in this study are provided in the Supplementary Data 5.

## Lipid staining

Oil red O staining was performed as described below. Oil red O (0.5%, Fujifilm Wako, 154-02072) in 100% isopropanol was made as a stock solution and mixed by vortexing. Guts were dissected and fixed with 4% paraformaldehyde (Nacalai Tesque, 09154-85) for 20 min. The guts were washed with 1× PBS twice. The oil red O stock solution was freshly mixed with Milli-Q water at a ratio of 3:2 by vortexing and filtrated using a 0.45 µm filter. The gut samples were incubated with 500 µL of fresh oil red O solution and incubated for 30 min. After washing with Milli-Q water twice, the guts were mounted in 80% glycerol. Images were obtained by a Leica MZ10F and a Zeiss Axio Virt.A1.

LipidTOX staining was performed as described below. Guts were dissected and fixed with 4% paraformaldehyde (Nacalai Tesque, 09154-85) for 20 min. The guts were washed with PBST (0.1% Triton X-100) three times and incubated with LipidTOX Deep Red neutral lipid stain (Invitrogen, H34477, 1:250) for two hours. After washing with PBST (0.1% Triton X-100), the guts were mounted in 80% glycerol. Images were obtained by a Zeiss Axio Virt.A1.

## Single-cell transcriptome analysis

Guts of anaesthetised adult female flies were dissected in 1 × PBS. The proventriculus and hindgut were carefully removed. The dissected

midguts were placed in ice-cold Schneider's medium (without FBS or penicillin/streptomycin). Collected midguts (100 midguts/sample) were washed briefly with PBS and transferred to a 1.5 mL Eppendorf tube containing 5 mg/mL elastase/PBS solution (SIGMA, E0258). The midgut solution was incubated on a shaker at 27 °C and 1000 rpm for 25 min with pipetting (40 times) every five minutes. Pipette tips were precoated with elastase solution to prevent the midgut from sticking to them. The dissociation reaction was stopped by adding 400 μL of Schneider's medium. The cell suspension was passed through a 100 μm cell strainer. The flowthrough was centrifuged at $600 \times g$ for 20 min, and the supernatant was discarded. The cell pellet was resuspended in 100 μL of Schneider's medium. Cell number and viability were assessed by trypan blue staining using a haemocytometer. The samples were then sent to Genble, Inc. Library preparation, sequencing, and data processing were performed as follows.

A single-cell suspension was obtained by filtering the cell suspension with a 40 μm cell strainer. Isolated single cells were loaded onto the 10x Chromium Controller (10x Genomics). Single-cell cDNA synthesis, amplification and sequencing library creation were performed by using the Single Cell 3′ Library kit v3.1 following the manufacturer's protocol. The libraries were sequenced on a NovaSeq 6000 (Illumina) platform with the following sequencing parameters: 28 bp read 1, 8 bp index 1, 91 bp read 2. Sequenced reads were subjected to demultiplexing, alignment, barcode counting, UMI counting, and filtering using Cell Ranger v.5.0.1. The *Drosophila melanogaster* genome (BDGP6.28) was used as a reference to align reads. Subsequent analysis was performed in R (version 3.6.0) using the package Seurat (version 3.1.0). Genes expressed in fewer than three cells were removed. The cells that expressed less than 200 genes or that expressed more than 3,000 genes were removed. We also removed the cells that had more than 25% of mitochondrial-associated genes among their expressed genes and more than 15,000 UMI counts. The UMI counts were log normalised (scale factor = 10,000). We regressed out the S scores, G2/M scores, and the percentage of mitochondrial-associated genes during data scaling. Dimensionality reduction was performed using principal component analysis (PCA). The first 50 principal components were used to cluster the cells based on the shared nearest neighbours (SNN) algorithm and visualised in two dimensions by uniform manifold approximation and projection (UMAP). Differential gene expression analysis was performed using MAST[78] in the Seurat package with min.pct at 0.2. The dataset has been deposited in the Gene Expression Omnibus database under accession code GSE198149.

### Heatmap, PCA, and dot plot analyses for transcriptome datasets

For the heatmap analysis and the PCA, DEGs expressed in more than two clusters in cells classified as ISC or EB were assessed in the progenitor cell cluster. DEGs expressed in more than six clusters in cells classified as aEC, mEC, copper cell/iron cell, pEC, or LFC were assessed in the enterocyte cluster. DEGs expressed in more than four clusters in cells classified as EE were assessed in the enteroendocrine cell cluster. PCA was performed using the mixOmics package (3.14)[58]. A dot plot was generated using the ggplot2 package (3.3.5).

### Statistical analysis

Statistical analysis was performed using GraphPad Prism 8/9. The sample numbers were determined empirically. All data points were biological, not technical, replicates. A two-tailed Student's *t* test was used to test between samples. One-way ANOVA with Holm-Šídák's multiple comparison test was used to test among groups. One-way ANOVA with Dunnett's multiple comparison test was used to test samples against controls. All experimental results were tested at least twice to confirm their reproducibility. Bar graphs are drawn as the mean and SEM. OASIS 2 was used to perform log-rank test for lifespan analysis[79]. Cox PH analysis was performed using R[80]. The Cox Proportional-Hazards Model ('coxph') function of the "Survival" package[81] was used to fit the model and the 'Anova' function of the "car" package[82] to assess lifespan differences between diets and timings of dietary treatment or between diets and genotype, as well as diet by timing interactions or diet by genotype interactions. The code is in a Supplementary Data 6.

### Reporting summary

Further information on research design is available in the Nature Portfolio Reporting Summary linked to this article.

## Data availability

All relevant data are available from the authors upon request. The NGS data generated in this study have been deposited in the DDBJ and GEO under accession codes DRA013585, DRA013644, and GSE198149. The genome datasets used in this study was BDGP6.22 for RNAseq analysis the guts of young or old flies upon 24 hr Met restriction, BDGP6 for 3′ RNAseq analysis of guts after 6 hours, 2 weeks, or 4 weeks of Met restriction, BDGP6.28 for single-cell transcriptome analysis. The metabolite analysis data generated in this study have been deposited in the DDBJ MetaboBank under accession codes MTBKS226, MTBKS227, MTBKS228, MTBKS229, MTBKS230, MTBKS231 [https://ddbj.nig.ac.jp/public/metabobank/study/]. Source data are provided with this paper.

## Code availability

No custom codes were used during this study.

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

## Acknowledgements

The authors would like to acknowledge Kyoto Stock Center, National Institute of Genetics, Vienna Drosophila Resource Center, and Bloomington Drosophila Stock Center for reagents. We thank Takao Fujisawa, Yoriko Akuzawa, and all the members of Miura's laboratory and Obata's laboratory for technical assistance and critical advice. We thank Sa Kan Yoo and Scott Pletcher for the critical reading of the manuscript. This work was supported by AMED-PRIME to F.O. under Grant Number JP20gm6310011 and by AMED-Project for Elucidating and Controlling Mechanisms of Aging and Longevity to M.M under Grant Number JP21gm5010001. This work was also supported by grants from the Japan Society for the Promotion of Science to F.O. under Grant Number 19H03367, 20H05726, and 22H02769, and to M.M. under Grant Numbers 16H06385, 21H04774, 21K19206, and 23H04766.

## Author contributions

M.M. and F.O. conceived the project. H.K., J.K., H.A., R. O., T. O. and F.O. performed the experiments and analysed the data. J.J and M. P. performed a statistical analysis of lifespan curves. H.K., M.M. and F.O. wrote the initial manuscript. M.M. supervised the study. All authors approved the final manuscript.

## Competing interests

The authors declare no competing interests.
