## [Peer Review File · Nature Communications]

REVIEWER COMMENTS

Reviewer #1 (Remarks to the Author):

In this manuscript, Obata et al. demonstrate in *Drosophila* that MetR during the first four weeks, but not in later stages, of adult life can efficiently extend lifespan. They demonstrate that expression of multiple longevity genes is upregulated by MetR and remains for two weeks after cessation of MetR. They further demonstrate that MsrA is required for the lifespan extension by early-life MetR. Although the findings are potentially interesting, I have several major and minor concerns that should be addressed before the manuscript can be accepted for publication.

1. The authors use 1.6 mM Methionine in the control diet. Comparing levels of methionine in chemically defined food to a standard sugar-yeast diet established that 1 mM methionine corresponds to the level of methionine in a regular diet (Parkhitko et al. PNAS 2022). Further elevation of methionine in the food would activate the transsulfuration pathway and potentially misinterpret the results of experiments. In several experiments, the authors used 1 mM methionine that would mitigate this concern.

2. The authors performed lifespan experiments with decreased concentration of all amino acids by 40%. It has been recently shown that in the genetic model of MetR the lifespan can be increased without decreasing the concentration of all amino acids (Parkhitko et al. PNAS 2022). Genetic model of MetR also significantly decreased the levels of methionine sulfoxide. The authors used a different diet than one that was used by Lee et al Nat Comm 2014. It is absolutely critical to use a diet with 100% of all amino acids as a control. Would MetR prolong lifespan on both diets? It would significantly increase the translational potential of the paper.

3. It has been recently shown that MsrA is not required for the beneficial metabolic effects of MetR in mice (Thyne et al. Scientific Reports 2022). This paper should be cited in the manuscript. It is possible that the unnatural experimental conditions (higher concentration of methionine and lowered concentration of all amino acids) can cause the discrepancy between fly and mouse data.

4. This part is written very unclearly:

“Absolute level of MetSO is ten times more

130 than Met, suggesting huge amount of Met is physiologically damaged (Fig. 1m). Upon

131 MetR, the level of MetSO was decreased (resulted in 0.227%), suggesting the additional

132 mechanism to downregulate the metabolite (Fig. 1m). It is noteworthy that other AAs

133 were also affected, however, Met level was decreased most strikingly (Supplementary
134 Fig. 1d). We do not know the mechanisms by which other AAs are increased or
135 decreased (Supplementary Fig. 1d).”

5. The authors used the long-rank test to compare the effect of early MetR and late MetR in WDah flies (Figures 2d and e). They propose that the effect with the late stage MetR is much smaller. I don't think the statistical analysis that was used here is correct. I believe the authors should use the Cox Proportional Hazards (CPH) analysis to compare lifespan effects between two pairwise groups. Same in Figure 6b,c.

6. It is unclear why the authors decided to switch from methionine restriction in lifespan studies to methionine starvation in transcriptome studies. It is well known in both *Drosophila* and mice that there is a sweet spot for methionine restriction to extend lifespan. Methionine restriction is beneficial as reflected by lifespan extension; methionine starvation – would dramatically shorten lifespan. Accordingly, the transcriptional response under methionine starvation would be dramatically different as under methionine restriction. Later, the authors come back to MetR to follow the transcriptional changes responsible for the early onset MetR.

7. The authors use the MsrAEY05753 P-element insertion line to demonstrate that MsrA is required for lifespan extension by early onset MetR. The authors should demonstrate by qRT-PCR that this line has reduced/no expression of MsrA and that the function of MsrA is compromised in this line by applying oxidative stress. The P element lines are often mismatched.

8. The authors should plot the lifespans for wildtype and MsrA mutant flies under control and MetR conditions on the same graph (especially considering that they were backcrossed at the same genetic background). It seems to me that MsrA mutant flies are long-lived compared to wild-type flies. If it is right, the interpretation would be completely different.

9. The authors compare lifespans between wildtype and MsrA mutant flies that are also different by the presence of mini w gene that is known to regulate lifespan and can further complicate the interpretation of this experiment.

10. Does lack of MsrA also prevent a positive effect of early onset MetR on climbing?

11. The authors use female flies though the paper because MetR does not have a significant effect in males. It would be critical to use males in parallel as a control. Would MsrA expression is upregulated in males without lifespan extension?

12. Does the observed effect in females simply depends on reproduction? Testing this in OvoD mutant flies will answer this question.

13. The language of the manuscript can be significantly improved.

Reviewer #2 (Remarks to the Author):

In this study, Obata et al investigated methionine-restriction (MetR) impacts on *Drosophila* longevity and found that MetR in young adult significantly extends lifespan. They also revealed that MsrA, a gene encoding a methionine sulfoxide reductase to diminish oxidatively-damaged Met, is upregulated by MetR and is essential for MetR-extended longevity. Using single-cell RNAseq, the authors indicated MsrA expression in ECs of adult anterior midgut that might contribute to lifespan extension under metR. As compared to numerous previous findings of MetR as well as MsrA in lifespan regulation in *Drosophila*, it is interesting to see predominant lifespan-extending roles of MetR in the first 4 weeks of adult in this study. The authors have also performed multi-omics and indicated quite a few interesting metabolic and stress-associated genes that are potentially involved in MetR. However, authors failed to provide novel insights into the mechanisms of early-adult MetR, including the specific organs contributing lifespan extension, key biological processes in these organs associated with MetR, and the differential roles of different methionine metabolites like MetSO, Cysta/Cys and so on.

Thus, I recommend a major revision and hope the authors address the following comments prior to publication in Nature Communication.

Major comments:

1. If the authors put Fig. 6B and 6C together, it is straightforward that MsrA mutation decreases longevity of control flies without MetR. The results are against their statement of “no apparent phenotype in a MsrA null mutant”. Thus, the authors need to confirm whether MsrA is required for general longevity regulation or MetR-associated lifespan extension alone.

2. Since the authors demonstrated MsrA expression in the fat body, midgut ECs, and maybe some EEs (from DRSC snRNAseq data: www.flyrnai.org/scRNA/gut/), why they did not test the cell-type-specific functions of MsrA in lifespan regulation? It will be important to dissect the key organs/tissues whereby MsrA and methionine metabolism sufficiently affect systemic longevity in the context of MetR.

3. The authors have performed multi-omics in this study and, importantly, have observed a lot of interesting genes that are differentially regulated. It is a pity for them to stop on only correlations between these genes and MetR. I would recommend them to further investigate and CONFIRM, at least, ONE of the biological processes in the midgut as an essential regulator of metR-induced lifespan extension in early adult stage. For example, are lipid composition, synthesis, degradation, or transport changed in the ECs? Is lipid metabolic perturbation sufficient to modulate MetR-associated lifespan extension? Immune response to virus or bacteria? The composition of different types of gut cells? The gut hormones like CCHA2 and its associated physiological impacts? ROS? Mitochondria? Otherwise, the novelty of this study is not significant enough for publication according to the published results of MetR and MsrA.

4. The same issues for MsrA in the fat body.

5. Methionine could be converted into MetSO as well as Cysta/Cys. In addition to Met/MetSO, the ratio of Met/Cysta is also blunted in aged flies in Fig. 3 in this study. The authors have previously revealed the roles of SAM/SAH, the upstream of Cysta/Cys, in lifespan modulation, it will be required to exclude the Cysta/Cys branch or examine potential crosstalk between MetSO and Cysta/Cys in methionine metabolism during early-adult MetR. The authors could easily knock down CBS and perform PPG administration to perturb Cysta/Cys metabolism in both gut and fat body in the context of MsrA knockdown.

Minor comments:

6. MsrA has been shown to catalyze multiple targets beside MetSO (Chung et al., 2010; Guan et al., 2017), is it possible to check the effects of other targets of MsrA in MetR? Like DMS?

References:

Chung, H., Kim, A.K., Jung, S.A., Kim, S.W., Yu, K., and Lee, J.H. (2010). The *Drosophila* homolog of methionine sulfoxide reductase A extends lifespan and increases nuclear localization of FOXO. *FEBS Lett* 584, 3609-3614. [10.1016/j.febslet.2010.07.033](https://doi.org/10.1016/j.febslet.2010.07.033).

Guan, X.L., Wu, P.F., Wang, S., Zhang, J.J., Shen, Z.C., Luo, H., Chen, H., Long, L.H., Chen, J.G., and Wang, F. (2017). Dimethyl sulfide protects against oxidative stress and extends lifespan via a methionine sulfoxide reductase A-dependent catalytic mechanism. *Aging Cell* 16, 226-236. [10.1111/acer.12546](https://doi.org/10.1111/acer.12546).

Reviewer #1 (Remarks to the Author):

In this manuscript, Obata et al. demonstrate in Drosophila that MetR during the first four weeks, but not in later stages, of adult life can efficiently extend lifespan. They demonstrate that expression of multiple longevity genes is upregulated by MetR and remains for two weeks after cessation of MetR. They further demonstrate that MsrA is required for the lifespan extension by early-life MetR. Although the findings are potentially interesting, I have several major and minor concerns that should be addressed before the manuscript can be accepted for publication.

1-1. The authors use 1.6 mM Methionine in the control diet. Comparing levels of methionine in chemically defined food to a standard sugar-yeast diet established that 1 mM methionine corresponds to the level of methionine in a regular diet (Parkhitko et al. PNAS 2022). Further elevation of methionine in the food would activate the transsulfuration pathway and potentially misinterpret the results of experiments. In several experiments, the authors used 1 mM methionine that would mitigate this concern.

>We thank the reviewer for pointing this out. In these experiments, we needed to use 1.6 mM Met (40% of the original 4 mM Met), which is based on studies that match the AA balance of the diet to the Drosophila exome (Piper et al., Cell Metabolism, 2017). If we decrease further the amount of Met, then this amino acid would become (relatively) insufficient, leading to misinterpretation of the result. As the reviewer mentioned, for key data, we also used 1 mM Met to confirm the result. Basically, we observed the similar phenotypes using this concentration.

We also note that the amino acid levels in fly diets depend on how much and what quality of yeast is used. We have quantified the amino acids in our standard yeast-based diet and found that it contains approximately 2 mM of Met. Interestingly, the original report of the synthetic diet, that we have used in this study, shows that all nutrients (and drugs) are up to 10-times more biologically available than those found in yeast-based foods (Piper et al *Nature Methods*, 2014). Thus, the biological availability of nutrients is heavily determined by nutritional context making it impossible to directly compare absolute amounts of nutrients between diets and deduce their effects on the consumer's physiology.

1-2. The authors performed lifespan experiments with decreased concentration of all amino acids by 40%. It has been recently shown that in the genetic model of MetR the lifespan can be increased without decreasing the concentration of all amino acids (Parkhitko et al. PNAS 2022). Genetic model of MetR also significantly decreased the levels of methionine sulfoxide. The authors used a different diet than one that was used by Lee et al Nat Comm 2014. It is absolutely critical to use a diet with 100% of all amino acids as a control. Would MetR prolong lifespan on both diets? It would significantly increase the translational potential of the paper.

> The revised manuscript has new figure 2j which contains five survival curves. In this experiment, we used 100 % AA (2.5 mM Met) and decreased the level of Met to either 10% (0.25 mM) or 4% (0.16 mM, an equivalent of 10% Met when 40% AA is used) throughout the life or only at the early adulthood. This analysis clearly demonstrated that 10% Met in 100% AA background can extend lifespan of female Canton-S, while 4% Met cannot. Intriguingly, 4% Met also fully extended lifespan when flies were flipped back to the 100% AA diet in the middle age. This striking observation suggested that 1) decreasing all AA into 40% is not mandatory for MetR-longevity and 2) For the lifetime MetR (but not for early MetR), 4% Met is too low when 100% AA is used. These data highlight that AA requirement is context dependent. We thank the reviewer for letting us find this striking observation that even a harsher decrease in Met levels during early adulthood is beneficial, increasing the translational potential of the study. We also added discussion that the genetic model of MetR leads to significant decrease in MetSO levels (Parkhitko et al., PNAS, 2022), supporting the generality of our finding.

Fig.2j, Lifespans of female Canton-S flies fed a methionine-restricted (10% or 4%) diet compared to 100% AA during their early life or whole life. Sample sizes (n) are shown in the figure. For the statistics, a log-rank test was used to compare with the control.

(Result: p.8) A recent study has shown that overexpression of bacteria-derived methioninase in *Drosophila* can break down internal Met and increase lifespan without decreasing the level of other amino acids³⁹. To assess if we observed extended lifespan because of reduced methionine or via an interaction between reduced methionine and other amino acids that we modified, we analysed lifespan of female Canton-S flies fed with holidic media containing 100% AA (containing 2.5 mM Met) with either 10% (0.25 mM Met) or 4% (0.16 mM Met, which is equivalent to 10% Met when 40% AA is used) of Met levels throughout life or only in early life. We found that both early and lifelong restriction of Met to 10% in 100% AA background can extend lifespan, but lifelong restriction of Met to 4% cannot (Fig. 2j). Intriguingly, limiting 4% Met restriction to early life only fully extended lifespan (Fig. 2j). This striking observation suggested that 1) decreasing all AA into 40% is not mandatory for MetR-longevity and 2) harsher MetR can extend lifespan when restricted to early adulthood.

(Discussion: p.19) In the genetic model of MetR, the level of MetSO is also strongly decreased, although the direct contribution of MsrA to this phenotype was not tested³⁹.

1-3. It has been recently shown that MsrA is not required for the beneficial metabolic effects of MetR in mice (Thyne et al. Scientific Reports 2022). This paper should be cited in the manuscript. It is possible that the unnatural experimental conditions (higher concentration of methionine and lowered concentration of all amino acids) can cause the discrepancy between fly and mouse data.

>We cannot agree with the reviewer that there is the discrepancy. Thyne et al. measures only metabolic phenotypes while we analysed longevity. These two phenotypes are not necessarily correlated. Indeed, Thyne et al. wrote this in the last paragraph of the paper. "*While our studies investigated these interactions and effects [on Metabolic function] in adult mice, it remains an open question as to their long-term effects on longevity and health span.*" We agreed with what the authors discussed here and thus cited this paper in the revised manuscript.

The outcome of dietary manipulation reasonably varies by contexts. As discussed above in our comment 1-2, our MetR covers a range of dietary manipulations: 0.15 mM-0.25 mM Met in 40% AA to 100% AA (containing 1.6 mM to 2.5 mM Met) can increase the fly lifespan. We also showed that three different control strains robustly induced the phenotype. These data sufficiently deny the possibility that our experimental condition is unnatural.

(Discussion; p.19) In contrast, MsrA was reported to be nonessential for metabolic benefit of MetR in mice⁶⁷. Whether MsrA is required for lifespan extension in other contexts in various organisms including mammals needs to be tested in the future.

1-4. This part is written very unclearly:

"Absolute level of MetSO is ten times more than Met, suggesting huge amount of Met is physiologically damaged (Fig. 1m). Upon MetR, the level of MetSO was decreased (resulted in 0.227%), suggesting the additional mechanism to downregulate the metabolite (Fig. 1m). It is noteworthy that other AAs were also affected, however, Met level was decreased most strikingly (Supplementary Fig. 1d). We do not know the mechanisms by which other AAs are increased or decreased (Supplementary Fig. 1d)."

> We thank the reviewer for pointing out here. We revised the text to clarify the sentences as below. The AA profile of female flies during the MetR is not directly related to the main conclusion of this study. However, we believe it is vital not to make a whole study oversimplified by omitting unclear data.

(Result; p.6) "In contrast, the level of MetSO was sharply downregulated to 0.227% of the control upon MetR. This resulted in a 28-fold decrease in MetSO when compared to the decrease in internal Met, suggesting the presence of an active mechanism to downregulate MetSO levels during MetR. Considering that the absolute level of MetSO in control animals was ten times greater than that of Met, a large amount of Met was physiologically damaged (Fig. 1m).

We also noticed that other AA levels were affected during MetR. Threonine, asparagine, glutamine, and glycine were increased, and leucine, phenylalanine, tryptophan, and tyrosine were decreased (Supplementary Fig. 1f). Although we do not know the mechanisms by which other AAs are increased or decreased, methionine metabolism is coupled to these amino acids via one carbon metabolism or mitochondrial metabolism³⁶⁻³⁸."

1-5. The authors used the long-rank test to compare the effect of early MetR and late MetR in WDah flies (Figures 2d and e). They propose that the effect with the late stage MetR is much smaller. I don't think the statistical analysis that was used here is correct. I believe the authors should use the Cox Proportional Hazards (CPH) analysis to compare lifespan effects between two pairwise groups. Same in Figure 6b,c.

>To quantitatively discuss the contribution of fly age, as the reviewer suggested, we have now worked together with Dr. Matthew Piper's lab to do CPH analysis for Fig. 2b-e and new Fig. 6b,h. The analysis identified there is indeed a statistical significance for "timing of dietary manipulation" for w^{iso31} female, but not for w^{Dah} . Although this would not influence our conclusion, we weakened the tone.

(Result; p.7) Analysing the data using cox proportional hazards revealed that the timing of methionine restriction changed its effects on lifespan; early MetR extended lifespan relative to non-restricted controls, while MetR later in life did not (Fig. 2b, c;

Supplementary Table 1). Interestingly, when we used the same protocol on w^{Dah} females, both early and late MetR had similar effects in that they both extended lifespan, although early MetR again had a stronger effect on median lifespan (14.5% increase) than later life MetR (9.7% increase) (Fig 2.d, e; Supplementary Table 1).

1-6. It is unclear why the authors decided to switch from methionine restriction in lifespan studies to methionine starvation in transcriptome studies. It is well known in both *Drosophila* and mice that there is a sweet spot for methionine restriction to extend lifespan. Methionine restriction is beneficial as reflected by lifespan extension; methionine starvation – would dramatically shorten lifespan. Accordingly, the transcriptional response under methionine starvation would be dramatically different as under methionine restriction. Later, the authors come back to MetR to follow the transcriptional changes responsible for the early onset MetR.

>The purpose of this analysis was not to identify the mechanism for MetR-longevity, but to ask how the organismal response to the shortage of dietary Met differs between ages. Therefore, as we stated in the manuscript, we simply compared the diets with or without Met to maximise the response to this amino acid. One reason to select this was that, while the lifespan analysis is chronic/cumulative (~60 days accumulation of the dietary effect), the gene expression analysis is acute/single time frame. For such a snap-shot analysis, it should be better to use a stronger manipulation. This is especially effective to consider the effect of age since longer duration of the manipulation *per se* would lead to ageing of animals. As mentioned by the reviewer, we indeed came back to the transcriptome analysis for MetR diet and confirmed that *MsrA* induction was a key event. These data together must have complemented the weaknesses of each analysis and significantly strengthened the robustness of the finding.

1-7. The authors use the MsrAEY05753 P-element insertion line to demonstrate that *MsrA* is required for lifespan extension by early onset MetR. The authors should demonstrate by qRT-PCR that this line has reduced/no expression of *MsrA* and that the

function of MsrA is compromised in this line by applying oxidative stress. The P element lines are often mismatched.

>We have now added the qRT-PCR data in new Supplementary Fig. 4a. It showed undetectable level of *MsrA* expression (the primer set has been designed to detect the mRNA outside of the P-element insertion), confirming it is null mutant. For oxidative stress, we tested survival against H₂O₂ feeding. The *MsrA* mutant did not show decreased oxidative stress resistance (new Supplementary Fig. 4b). This might be due to the redundant function of MsrB. It is also possible that the increase of mild oxidative stress in the mutant flies induces a hormetic effect, reinforcing the stress resistance and extend lifespan. We added the data and interpretation in the revised manuscript

Supplementary Fig. 4a, Quantitative RT-PCR analysis of *MsrA* in the whole bodies of *w^{Dah}* or *MsrA^{EY}* flies. $n = 4$. For the statistics, a two-tailed Student's *t* test was used. **B**, Survivability of female flies of *w^{Dah}* or *MsrA^{EY}* upon 3% H₂O₂ treatment. Sample sizes (n) are shown in the figure. For the statistics, a log-rank test was used.

(Result; p.12) We confirmed by qRT-PCR that there was no detectable level of *MsrA* transcripts in the mutant (Supplementary Fig. 4a).

(Result; p.13) Surprisingly, the lifespan of the *MsrA^{EY}* female fly is longer than that of *w^{Dah}* on the control diet (Fig. 6b). This could be because there is an increase in mild oxidative stress in the mutant flies that induces a hormetic effect, reinforcing the stress

resistance and extending lifespan. Indeed, the mutant showed an increased resistance to hydrogen peroxide (Supplementary Fig. 4b).

1-8. The authors should plot the lifespans for wildtype and MsrA mutant flies under control and MetR conditions on the same graph (especially considering that they were backcrossed at the same genetic background). It seems to me that MsrA mutant flies are long-lived compared to wild-type flies. If it is right, the interpretation would be completely different.

>We revised the data to incorporate four lifespan curves into one graph (new Fig. 6b). Indeed, The *MsrA* mutant are not short-lived, but rather long-lived under the control diet. As pointed in our comment 1-7, we observed that the *MsrA* mutant had higher resistance to oxidative stress (new Supplementary Fig. 4b), suggesting that a mild increase of oxidative stress in the mutant might induce a possible hormetic effect, which reinforces stress resistance and extends lifespan. Indeed, we demonstrated that the *MsrA* mutant had only 30% increase of MetSO, which would not be strong enough to shorten the lifespan. In sharp contrast, MetR decreases MetSO/Met ratio very strongly, which is completely abolished in the *MsrA* mutant. Therefore, a decrease in MetSO by the induced *MsrA* by the MetR diet is beneficial (and extend lifespan), but a slight increase of MetSO by loss of function of MsrA in the control diet would not be detrimental. We added these data and interpretation to the revised manuscript.

Fig. 6b, Lifespans of female flies of w^{Dah} and $MsrA^{EY}$ fed with or without a methionine-restricted diet in early life. Sample sizes (n) are shown in the figure. For the statistics, a log-rank test was used.

(Result; p.13) Surprisingly, the lifespan of the $MsrA^{EY}$ female fly is longer than that of w^{Dah} on the control diet (Fig. 6b). This could be because there is an increase in mild oxidative stress in the mutant flies that induces a hormetic effect, reinforcing the stress resistance and extending lifespan. Indeed, the mutant showed an increased resistance to hydrogen peroxide (Supplementary Fig. 4b).

1-9. The authors compare lifespans between wildtype and $MsrA$ mutant flies that are also different by the presence of mini w gene that is known to regulate lifespan and can further complicate the interpretation of this experiment.

>Unfortunately, we have only the $MsrA$ null mutant with a mini w gene insertion, which is backcrossed to the control w^{Dah} strain. As we proved that the MetR-driven lifespan extension and many other phenotypes are similarly observed in Canton-S (w^+), w^{Dah} (w^-), and w^{iso31} (w^-), it is less likely that the presence of white gene could account for the $MsrA$ mutant phenotype. Indeed, a well-controlled lifespan experiment in the literature suggested that the white mutation did not influence the lifespan at all (Sasaki et al., Nat Metab, 2021). To be fair, in the revised manuscript, we added the following sentence honestly explaining the limitation of the study.

(Result: p.13) Note that, in this analysis, *MsrA^{EY}* mutant flies have an insertion of mini white gene, while the control flies do not. The dose of white could have affected the lifespan, although it cannot account for the change in response to MetR since both red and white-eyed flies have extended lifespan in response to MetR (Fig. 2a, b, d).

1-10. Does lack of MsrA also prevent a positive effect of early onset MetR on climbing?
>We performed the climbing assay. Unfortunately, we could not observe the significant increase of climbing ability by MetR in *w^{Dah}* control strain (new Supplementary Fig. 1e). *w^{Dah}* might have been too active to detect the improvement of climbing ability. We added the data in the revised manuscript. In our hindsight, we backcrossed the *MsrA* mutant with *w^{Dah}*, so we could not conclude whether the *MsrA* induction contributes to the improved climbing.

Supplementary Fig. 1e, Climbing abilities of female Canton-S or *w^{Dah}* flies fed with or without a methionine-restricted diet for four weeks.

(Result; p.6) The flies fed with the MetR diet for four weeks showed significantly improved climbing ability, although this phenotype was not observed in *w^{Dah}* female flies (Supplementary Fig. 1e).

1-11. The authors use female flies though the paper because MetR does not have a

significant effect in males. It would be critical to use males in parallel as a control.

Would *MsrA* expression is upregulated in males without lifespan extension?

1-12. Does the observed effect in females simply depends on reproduction? Testing this in *OvoD* mutant flies will answer this question.

>First, we tested the *MsrA* expression in males. Interestingly, *MsrA* is induced in males, suggesting the dietary response is not sex-biased (new Supplementary Fig. 3a-d). On the other hand, *ovo^{D1/+}* mutant could not increase their lifespan in response to MetR (new Fig. 2i). Even more intriguingly, *ovo^{D1/+}* mutant cannot increase *MsrA* expression upon MetR (new Supplementary Fig. 3e), suggesting that the *MsrA* induction in females depends on the reproductive capacity. These data together suggested that egg production and *MsrA* are two indispensable components to increase the lifespan by MetR. We added these data in the revised manuscript.

Fig. 2i, Lifespans of female *ovo^{D1/+}* flies fed with or without a methionine-restricted diet during early life. Sample sizes (*n*) are shown in the figure. For the statistics, a log-rank test was used.

Supplementary Fig. 3a-d, Quantitative RT–PCR analysis of *MsrA* expression levels in female guts (a), male guts (b), female abdomens (c) and male abdomens (d) of Canton-S flies fed with or without a methionine-restricted diet for three days. $n = 4$. For the statistics, a two-tailed Student's *t* test was used.

Supplementary Fig. 3e, Quantitative RT–PCR analysis of *MsrA* expression in female guts of *ovo*^{D1/+} fed with or without a methionine-restricted diet for three days. $n = 6$. For the statistics, a two-tailed Student's *t* test was used.

(Result; p.8)

We also tested the lifespan in *ovo*^{D1/+} mutant female flies, in which no egg production was observed. The mutant had a relatively longer lifespan than fertile females in the control diet, and *MetR* in the first four weeks did not increase female lifespan further, suggesting that reproduction is indispensable for lifespan extension during early *MetR* (Fig. 2i).

(Result; p.12) *MsrA* could be upregulated as early as three days of *MetR* in both the gut and abdomen of males and females (Supplementary Fig. 3a-d). The *ovo*^{D1/+} mutant females did not significantly increase *MsrA* expression upon *MetR* (Supplementary Fig. 3e), suggesting that *MsrA* induction in females depends on the reproductive capacity.

1-13. The language of the manuscript can be significantly improved.

>The authors struggled to write the manuscript very carefully. We realise that, for non-native speakers, it is particularly difficult to make perfect writing. Therefore, we engaged a professional language editing service, in which two experts edited the manuscript. For further improving the language, we have now used a second company. We hope this further improved the language.

Reviewer #2 (Remarks to the Author):

In this study, Obata et al investigated methionine-restriction (MetR) impacts on *Drosophila* longevity and found that MetR in young adult significantly extends lifespan. They also revealed that *MsrA*, a gene encoding a methionine sulfoxide reductase to diminish oxidatively-damaged Met, is upregulated by MetR and is essential for MetR-extended longevity. Using single-cell RNAseq, the authors indicated *MsrA* expression in ECs of adult anterior midgut that might contribute to lifespan extension under metR. As compared to numerous previous findings of MetR as well as *MsrA* in lifespan regulation in *Drosophila*, it is interesting to see predominant lifespan-extending roles of MetR in the first 4 weeks of adult in this study. The authors have also performed multi-omics and indicated quite a few interesting metabolic and stress-associated genes that are potentially involved in MetR. However, authors failed to provide novel insights into the mechanisms of early-adult MetR, including the specific organs contributing lifespan extension, key biological processes in these organs associated with MetR, and the differential roles of different methionine metabolites like MetSO, Cysta/Cys and so on.

Thus, I recommend a major revision and hope the authors address the following comments prior to publication in Nature Communication.

Major comments:

2-1. If the authors put Fig. 6B and 6C together, it is straightforward that *MsrA* mutation decreases longevity of control flies without MetR. The results are against their statement of “no apparent phenotype in a *MsrA* null mutant”. Thus, the authors need to confirm whether *MsrA* is required for general longevity regulation or MetR-associated lifespan extension alone.

> We thank the reviewer for pointing this out. Please see our comment 1-8, where we explained why *MsrA* mutant was rather long-lived in the control diet. We assume a possible hormetic effect.

2-2. Since the authors demonstrated MsrA expression in the fat body, midgut ECs, and maybe some EEs (from DRSC snRNAseq data: www.flyrnai.org/scRNA/gut/), why they did not test the cell-type-specific functions of MsrA in lifespan regulation? It will be important to dissect the key organs/tissues whereby MsrA and methionine metabolism sufficiently affect systemic longevity in the context of MetR.

>We have now tested tissue-specific *MsrA*-RNAi (in adult stage) in the gut, fat body, or neurons for the MetR-longevity. First, we screened which RNAi line worked well by quantifying the MetSO levels in the whole body. The knockdown of MsrA using three different RNAi stocks with a ubiquitous driver *da-Gal4* revealed that none of the RNAi is effective enough to increase the amount of MetSO in the control diet. However, *UAS-MsrA*-RNAi[GD] is the most effective line to block the MetR-dependent decrease in MetSO levels, although it is still milder than the *MsrA*[*EY*] null mutant (new Supplementary Fig. 5a). Using this RNAi line, we have knocked down *MsrA* in the gut (TIGS), fat body (WBFBS), or neurons (ElavGS) and tested whether it influenced the MetR-longevity. Interestingly, the only tissue that affects the longevity upon RU treatment is the fat body, where the knockdown increased the lifespan in the control diet (Ctrl EtOH vs Ctrl RU). This resulted in a decrease in the lifespan extension by early MetR (8.2% extension for RU486 vs 13.8% for EtOH). This seemed to resemble the phenotype of *MsrA*^{*EY*} mutant, although the magnitude of the phenotype is much smaller. In contrast, other two drivers did not affect the MetR-longevity at all. Therefore, the action of MsrA could be predominantly in the fat body. However, due to the insufficient loss of function, we could not completely conclude the tissue is the only place of MsrA to act. Further analysis using a conditional knock out with multiple Gal4 drivers or RNAi with higher efficiency is necessary to identify the responsible cell types.

Supplementary Fig. 5 Lifespan extension upon methionine restriction with tissue-specific *MsrA* knockdown.

a, Quantification of methionine sulfoxide in the whole bodies of female flies with *MsrA* knockdown using the *da-Gal4* driver upon methionine restriction. $n = 6$. For the statistics, one-way ANOVA with Holm-Šidák's multiple comparison test was used.

b-d, Lifespans of female flies with *MsrA* knockdown using *ElavGS* (brain driver, **b**), *TIGS* (gut driver, **c**), or *WBFGBS* (fat body driver, **d**) drivers fed with or without a methionine-restricted diet in early life. Sample sizes (n) are shown in the figure. For the statistics, a log-rank test was used. For the graph, the mean and SEM are shown. Data points indicate biological replicates.

(Result; p.13)

To identify the tissue responsible for the *MsrA*-driven lifespan extension by early MetR, we performed tissue-specific *MsrA*-RNAi. First, we screened which RNAi line worked well by quantifying the MetSO levels. Knock down of *MsrA* using three different RNAi stocks with a ubiquitous driver *da-Gal4* revealed that none of the stocks had increased MetSO in the control diet, indicating that *MsrA* knock down was inefficient

(Supplementary Fig. 5a). However, UAS-MsrA-RNAi^{GD} was still effective to block the MetR-dependent decrease in MetSO levels to some extent, although it was much milder than the MsrA^{EY} null mutant (Fig. 6g, Supplementary Fig. 5a). Using this RNAi line, we knocked down MsrA in the gut (using the TIGS driver), fat body (with the WFBGS driver), or neurons (using the ElavGS driver) and tested whether it influenced the lifespan extension by early MetR (Supplementary Fig. 5b-d). Interestingly, the only flies that have altered longevity upon transgene induction by RU486 treatment is the fat body knockdown (WFBGS>MsrA-RNAi), where the knock down increased the lifespan in the control diet (Ctrl EtOH vs Ctrl RU) (Supplementary Fig. 5d). This resulted in a decrease in the lifespan extension by early MetR (8.2% extension for RU486 vs. 13.8% for EtOH control). This blunted lifespan response to MetR seemed to resemble that of the MsrA^{EY} mutant (Fig. 6b). In contrast, the other two drivers did not affect the MetR-longevity at all. Therefore, the action of MsrA could be predominantly in the fat body. However, due to the inefficiency of RNAi, we could not conclude that the tissue is the only place in which MsrA acts. Further analysis using a conditional knockout with multiple Gal4 drivers or RNAi with higher efficiency will be necessary to identify the responsible tissue/cell types.

2-3. The authors have performed multi-omics in this study and, importantly, have observed a lot of interesting genes that are differentially regulated. It is a pity for them to step on only correlations between these genes and MetR. I would recommend them to further investigate and CONFIRM, at least, ONE of the biological processes in the midgut as an essential regulator of metR-induced lifespan extension in early adult stage. For example, are lipid composition, synthesis, degradation, or transport changed in the ECs? Is lipid metabolic perturbation sufficient to modulate MetR-associated lifespan extension? Immune response to virus or bacteria? The composition of different types of gut cells? The gut hormones like CChA2 and its associated physiological impacts? ROS? Mitochondria? Otherwise, the novelty of this study is not significant enough for publication according to the published results of MetR and MsrA.

> Indeed, due to the thorough description of the phenotype by multi-omics in this study, we have too many genes/pathways to test functionally. Thus, we focused on

one gene, *MsrA*, for further investigation of the function using both genetics and analytical chemistry, and we believe it is reasonable for one paper. Nevertheless, we agree with the reviewer that it is interesting to test other genes in order to discuss how these genes uniquely or commonly contribute to the MetR-longevity. During revision, we analysed two lipid-related genes, *Lsd-2* and *bmm*, which were robustly upregulated in the gut and fat body upon MetR. Lipid metabolism is known to have striking impact on lifespan extension in classical dietary restriction (DR) models (Katewa et al., Cell Metab, 2012, Luis et al., Cell Rep, 2016). DR promotes lipid turnover in the gut, leading to an increased starvation resistance and the concomitant longevity.

First, we performed lipid staining in the gut. Increased lipid signal in the gut is a hallmark of DR for starvation resistance and longevity (Luis et al., Cell Rep, 2016). As observed in DR, MetR increased neutral lipids in the gut of wild type flies (new supplementary Fig. 1a). Together with the increased gene expression of *Lsd-2* and *bmm*, this phenotype suggested that MetR promotes intestinal lipid turnover, phenocopying what happened during the conventional DR. However, when we knocked down these genes in the gut or fat body, the MetR-longevity was not blunted, suggesting that the altered lipid metabolism might not involve the lifespan regulation in our context (new Supplementary Fig. 6c-f).

To negate the possibility that knockdown efficiencies were too low, as was the case of *MsrA*, we also tested null mutant of *bmm* (new Fig. 6h, Supplementary Fig. 6a,b). In this mutant, neutral lipid was already abundant in the control diet, and mildly upregulated upon MetR, which is correlated well with the starvation resistance (new Supplementary Fig. 6a,b). Similar to the RNAi experiment, the lifespan extension by MetR was not inhibited in the *bmm* mutant at all (new Fig. 6h), further confirming that the altered lipid turnover in the gut has little contribution to the MetR-longevity. It is still possible that blocking other lipid metabolic genes or upstream transcription factors in the gut or in the other tissues alters the MetR longevity, as in DR (Katewa et al., Cell Metab, 2012, Luis et al., Cell Rep, 2016). This direction, together with the functional analysis of other genes we identified in the omics analyses, should be well beyond the scope of the current study and would be addressed in the future studies. We added these data and discussion in the revised manuscript.

Supplementary Fig. 1a, b, Oil red O staining of female guts of Canton-S flies fed with or without a methionine-restricted diet for nine days. Whole gut (a) or magnified view of the anterior midgut (b). Scale bar: 1 mm (a), or 100 μ m (b).

Supplementary Fig. 6 Contribution of lipid metabolism to lifespan extension upon methionine restriction.

a, Lipid staining of the female guts of *bmm*^{wt} and *bmm*¹ flies fed with or without a methionine-restricted diet for one week using LipixTOX. Scale bar: 1 mm. Arrowheads

indicate lipid accumulation. **b**, Survivability of female bmm^{wt} and bmm^1 flies upon complete starvation after feeding with or without a methionine-restricted diet for one week. Sample sizes (n) are shown in the figure. For the statistics, a log-rank test was used. **c, d**, Lifespans of female flies with knocked down bmm using WBFBGS (fat body driver, **c**) or TIGS (gut driver, **d**) fed with or without a methionine-restricted diet in early life. Sample sizes (n) are shown in the figure. For the statistics, a log-rank test was used. **e, f**, Lifespans of female flies with $Lsd-2$ knockdown using WBFBGS (fat body driver, **e**) or TIGS (gut driver, **f**) fed with or without a methionine-restricted diet in early life. Sample sizes (n) are shown in the figure. For the statistics, a log-rank test was used.

Fig. 6h, Lifespans of female flies of bmm^{WT} and bmm^1 fed with or without a methionine-restricted diet in early life. Sample sizes (n) are shown in the figure. For the statistics, a log-rank test was used.

(Result; p.5) It has been reported that DR increases starvation resistance via increased lipids in the gut (Luis et al., Cell Rep., 2016). As expected, we observed accumulation of lipid staining in the gut during MetR (Supplementary Fig. 1a, b).

(Result; p.14)

MetR promoted lipid accumulation in the gut of wild-type flies (Supplementary Fig. 1a, b) and upregulated the gene expression of $Lsd-2$ and bmm (Fig. 4d,h, Supplementary Fig. 2e). This phenotype implies that MetR promotes lipid turnover, phenocopying what happened during conventional DR^{28,52}. To test whether altered lipid metabolism

*contributes to early MetR-longevity, we analysed a *bmm*¹ mutant and its control animals with the same genetic background^{42,53}. In this mutant, neutral lipid in the gut was already abundant under the control diet, and were mildly upregulated upon MetR, which correlated well with the flies' enhanced starvation resistance (Supplementary Fig. 6a,b). The mutation did not compromise the lifespan extension by MetR (Fig. 6h, Supplementary Table 3), suggesting that the altered lipid signal in the gut contributed little to MetR-longevity. Similarly, knock down of either *Lsd-2* or *bmm* in the gut or in the fat body did not blunt the MetR-induced lifespan extension (Supplementary Fig. 6c-f). Although we cannot rule out the possibility that blocking other lipid metabolic genes or upstream transcription factors in the gut or in the other tissues could alter the MetR longevity, these data suggested that the altered lipid metabolism might not involve the lifespan regulation in this context.*

2-4. The same issues for MsrA in the fat body.

>Please see our responses to 2-2 and 2-3.

2-5. Methionine could be converted into MetSO as well as Cysta/Cys. In addition to Met/MetSO, the ration of Met/Cysta is also blunted in aged flies in Fig. 3 in this study. The authors have previously revealed the roles of SAM/SAH, the upstream of Cysta/Cys, in lifespan modulation, it will be required to exclude the Cysta/Cys branch or examine potential crosstalk between MetSO and Cysta/Cys in methionine metabolism during early-adult MetR. The authors could easily knock down CBS and perform PPG administration to perturb Cysta/Cys metabolism in both gut and fat body in the context of MsrA knockdown.

> We thank the reviewer for this great suggestion. Although the level of cystathionine, unlike MetSO, is indeed decreased in aged flies upon MetR (which suggests this branch may not be affected by ageing), it is possible that flux of Met to transsulfuration pathway (TSP) might be blunted. To test whether the TSP can contribute to the MetR-longevity, we fed PPG and measured lifespan. First, we tested various concentration of PPG and quantified the internal Met metabolites during MetR. Cystathionine is accumulated by PPG in a dose dependent manner. Met, SAM and SAH are decreased

by MetR regardless of the PPG. Importantly, MetSO is strongly decreased by MetR independently of PPG, suggesting that cystathionine, but not MetSO and the upstream Met metabolites, is specifically affected by PPG. Administration of PPG shortened lifespan, however MetR was still able to extend lifespan. This striking data suggested that TSP did not contribute to the MetR-longevity.

Supplementary Fig. 7 Transsulfuration pathway is not involved in lifespan extension upon methionine restriction.

a, Methionine metabolic and transsulfuration pathways, which can be inhibited by propargylglycine (PPG). **b-f**, Quantification of methionine metabolites and their oxidative products upon methionine restriction. $n = 3$. For the statistics, one-way ANOVA with Holm-Šidák's multiple comparison test was used. **g**, Lifespans of female Canton-S flies fed with or without a methionine-restricted diet supplemented with 0.5 mM PPG. Sample sizes (n) are shown in the figure. For the statistics, a log-rank test was used. For all graphs, the mean and SEM are shown. Data points indicate biological replicates.

(Result, p.15)

Methionine metabolism is coupled with cysteine metabolism through the transsulfuration pathway (TSP) (Supplementary Fig. S7a). To test whether TSP is related to MetR-longevity, we used the TSP inhibitor propargylglycine (PPG). First, we fed female flies with various concentrations of PPG during MetR and quantified the internal Met metabolites. As expected, cystathionine accumulated in response to PPG in a dose dependent manner, while the levels of Met, SAM, SAH and MetSO did not respond to the PPG concentration (Supplementary Fig. 7b-f). The administration of PPG shortened the female lifespan (Supplementary Fig. 7g). Strikingly, however, we still observed lifespan extension by early-adult MetR (Supplementary Fig. 7g). From these data, we concluded that TSP is not involved in the MetR-induced longevity.

Minor comments:

2-6. MsrA has been shown to catalyze multiple targets beside MetSO (Chung et al., 2010; Guan et al., 2017), is it possible to check the effects of other targets of MsrA in MetR? Like DMS?

> As far as we understand, DMS is enriched in marine algae and thus not included in the normal fly condition. Theoretically, increased *MsrA* during MetR should catalyze any targets other than MetSO, if present. Given that it is intriguing that MsrA is required for DMS-induced lifespan, we cited the paper and added the following discussion.

(Discussion; p.19) Interestingly, dimethyl sulfide (DMS) is reported to extend lifespan in C. elegans and in Drosophila in an MsrA-dependent manner⁶⁶.

References:

Chung, H., Kim, A.K., Jung, S.A., Kim, S.W., Yu, K., and Lee, J.H. (2010). The Drosophila homolog of methionine sulfoxide reductase A extends lifespan and increases nuclear localization of FOXO. FEBS Lett 584, 3609-3614. 10.1016/j.febslet.2010.07.033.
Guan, X.L., Wu, P.F., Wang, S., Zhang, J.J., Shen, Z.C., Luo, H., Chen, H., Long, L.H., Chen, J.G., and Wang, F. (2017). Dimethyl sulfide protects against oxidative stress and extends lifespan via a methionine sulfoxide reductase A-dependent catalytic mechanism. Aging Cell 16, 226-236. 10.1111/accel.12546.

REVIEWER COMMENTS

Reviewer #1 (Remarks to the Author):

The authors significantly revised the manuscript, addressed all the points, and added new data broadening and strengthening the scope of the manuscript. The new findings that 1. MetR extends lifespan in the presence of 100% of AA; 2. dependency on reproduction; 3. accumulation of lipids in the gut; 4. hormetic/beneficial role of MsrA loss under control conditions and the detrimental role of MsrA under MetR conditions; 5. the finding that the TSP pathway is not involved are very interesting.

I have several comments that should be addressed before publishing the manuscript:

1. The same journal previously published that MetR in the presence of 100% of AA does not extend lifespan in flies. The current work clearly demonstrates that lifelong MetR extends lifespan even in the presence of 100% of AA reflecting a similar finding with the genetic model of MetR. I believe it would be important to highlight this in the abstract as one of the critical findings.

2. The authors added new data on the tissue-specific role of MsrA using tissue-specific GeneSwitch lines. This new piece of data lacks important controls: the lifespans were not performed in parallel with control RNAi. RNAi expression by itself may affect lifespan in some of these drivers. Many of these drivers are leaky. It makes impossible to distinguish if the tissue-specific expression of MsrA RNAi does not affect lifespan because it does not play a role in this tissue or because leaky expression of MsrA RNAi is strong enough to downregulate MsrA even in the absence of RU486 inducer. In my opinion, removing this piece would not affect the overall impact of this manuscript, while having this data in its current state may lead to misinterpretation.

3. The central part of the manuscript is using chemically-defined food for manipulating methionine levels. It is obligatory to include a detailed protocol on how this food was prepared, how long it was stored, catalog numbers for all food components, etc. Referencing another paper on this is not sufficient. I would suggest making 1) a supplementary table with all components, their catalog numbers, how the stocks solutions were prepared, whether they were filtered and/or protected from light, their pH, how long they were stored and at which temperature; 2) the detailed step-by-step protocol on how different components were prepared, mixed, at which temperature, their pH, storage time, etc. 3) making a supplementary video on how the food was prepared.

It is especially critical as the same journal previously published that MetR in the presence of 100% of AA does not extend lifespan in flies and the authors also used a different type of chemically defined diet. It

would be important to compare the diet compositions and the protocols for how the food was prepared.

4. R code for the CPH analysis used for this work should be included as a supplementary file.

5. The authors have information on the number of flies used for each lifespan. It would be helpful to add in the material and methods whether the lifespans were done once using multiple vials in parallel or repeated twice separated in time. If they were separated in time whether the batch covariate was included in the statistical analysis.

6. It would be helpful to demonstrate as a supplementary figure a comparison of lifespans at least for one wild-type strain used in the study for flies kept on a standard SY diet versus flies kept on chemically defined diet. Using chemically defined diet may decrease the lifespan of control flies and it is important to consider for the data interpretation.

7. It should be discussed in the main text how food calorie content (absolute and in %) differs under diets with different concentrations of methionine.

Reviewer #2 (Remarks to the Author):

The authors have addressed several comments of mine in the revised manuscript, such as the general aging effects of MsrA-null mutation and the differential impacts of methionine-associated metabolites. However, I still have concerns about the mechanisms of MetR. They failed to figure out the major target tissues for “methionine memory” or, alternatively, the biological processes that are regulated by “methionine memory” to modulate aging process (they excluded bmm/Lsd2-associated lipid metabolism though). This work did not provide sufficient significances or novelties into the field without these mechanisms as compared to previously-published studies such as MetR and mitophagy (PMID: 31850341), cell turnover (33902813), glucose sensing (32821821), and so on. I am not convinced by the results in Fig. S5 as well. Therefore, I still recommend the authors to address the following issues:

1. They checked knockdown efficiencies of different RNAi lines using Da-GS in Fig. S5A. Does Da-GS-induced MsrA knockdown phenocopy MsrA-null mutation regarding MetR and lifespan extension? The observations were all obtained from MsrA-null mutation, the single genetic evidence in this study.

2. In Fig. S5, MetR extended median lifespan of elav-GS control by ~50%, why only 5% extension in TI-GS and 10% in WB-FB-GS were found? They should repeat these experiments or confirm them ESPECIALLY in the GUT using other GS or Gal4-Gal80TS system. Their results have indicated gut MsrA as a critical regulator. It could be the global effects, not in single organs, of MsrA, but the authors need to validate with solid results.

Reviewer #1 (Remarks to the Author):

The authors significantly revised the manuscript, addressed all the points, and added new data broadening and strengthening the scope of the manuscript. The new findings that 1. MetR extends lifespan in the presence of 100% of AA; 2. dependency on reproduction; 3. accumulation of lipids in the gut; 4. hormetic/beneficial role of MsrA loss under control conditions and the detrimental role of MsrA under MetR conditions; 5. the finding that the TSP pathway is not involved are very interesting.

I have several comments that should be addressed before publishing the manuscript:

1. The same journal previously published that MetR in the presence of 100% of AA does not extend lifespan in flies. The current work clearly demonstrates that lifelong MetR extends lifespan even in the presence of 100% of AA reflecting a similar finding with the genetic model of MetR. I believe it would be important to highlight this in the abstract as one of the critical findings.

Response: We deeply thank the reviewer for this and all other comments to our manuscript, which significantly improved the impact of the study. We now include this statement and modify the abstract accordingly (so that it fits the word limit of the journal).

(Abstract p.2) Methionine restriction (MetR) extends lifespan in various organisms, but its mechanistic understanding remains incomplete. Whether MetR during a specific period of adulthood increases lifespan is not shown. In *Drosophila*, MetR is reported to extend lifespan only when amino acid levels are low. Here, by using an exome-matched holidic medium, we show that decreasing Met levels to 10% extends *Drosophila* lifespan with or without decreasing total amino acid levels. MetR during the first four weeks of adult life robustly extends lifespan. MetR induces the expression of *Methionine sulfoxide reductase A (MsrA)* in young flies, which reduces the oxidatively-damaged Met. *MsrA* induction is *foxo*-dependent and persists for two weeks after cessation of the MetR diet. Loss of *MsrA* attenuates lifespan extension by early-adult MetR. Our study highlights the age-dependency of the organismal response to specific nutrient and suggests that nutrient restriction at a particular period of life is sufficient for healthspan extension.

2. The authors added new data on the tissue-specific role of MsrA using tissue-specific GeneSwitch lines. This new piece of data lacks important controls: the lifespans were not performed in parallel with control RNAi. RNAi expression by itself may affect lifespan in some of these drivers. Many of these drivers are leaky. It makes impossible to distinguish if the tissue-specific expression of MsrA RNAi does not affect lifespan because it does not play a role in this tissue or because leaky expression of MsrA RNAi is strong enough to downregulate MsrA even in the absence of RU486 inducer. In my opinion, removing this piece would not affect the overall impact of this manuscript, while having this data in its current state may lead to misinterpretation.

Response: We agree with the reviewer that these data may not be conclusive and therefore we removed entire Fig. S5. Fig. S6c-f have also been removed as they had the same problem.

3. The central part of the manuscript is using chemically-defined food for manipulating methionine levels. It is obligatory to include a detailed protocol on how this food was prepared, how long it was stored, catalog numbers for all food components, etc. Referencing another paper on this is not sufficient. I would suggest making 1) a supplementary table with all components, their catalog numbers, how the stocks solutions were prepared, whether they were filtered and/or protected from light, their pH, how long they were stored and at which temperature; 2) the detailed step-by-step protocol on how different components were prepared, mixed, at which temperature, their pH, storage time, etc. 3) making a supplementary video on how the food was prepared. It is especially critical as the same journal previously published that MetR in the presence of 100% of AA does not extend lifespan in flies and the authors also used a different type of chemically defined diet. It would be important to compare the diet compositions and the protocols for how the food was prepared.

Response: We now added Supplementary Data 4 which have all the ingredients, their catalog numbers, and procedures to make stock solutions and the holidic medium. The protocol for medium preparation *per se* is not original and has been well described previously (Piper et al., Nat Methods, 2014 ,Piper et al., Cell Metab., 2017). As the step-by-step protocol is also deposited online (<https://www.protocols.io/view/holidic-media-hm-preparation-bp2l6815gqe5/v1>), we would not repeat it in this study. For video recording, we consider to publish it in future by other ways such as JoVE.

4. R code for the CPH analysis used for this work should be included as a supplementary file.

Response: We now added supplementary file 5.

5. The authors have information on the number of flies used for each lifespan. It would be helpful to add in the material and methods whether the lifespans were done once using multiple vials in parallel or repeated twice separated in time. If they were separated in time whether the batch covariate was included in the statistical analysis.

Response: We have done the lifespans from multiple vials in parallel.

(Methods p.24) For each lifespan curve, at least six vials were used in parallel to minimise inter-vial variation.

6. It would be helpful to demonstrate as a supplementary figure a comparison of lifespans at least for one wild-type strain used in the study for flies kept on a standard SY diet versus flies kept on chemically defined diet. Using chemically defined diet may decrease the lifespan of control flies and it is important to consider for the data interpretation.

Response: Unfortunately, we have never straightforwardly compared the lifespan of control flies in our SY diet and the holidic medium. However, it has been shown that lifespan under holidic diet is similar to that under a SY diet (Piper et al., Nat Methods, 2014). In our experimental setting, holidic medium does not shorten lifespan, either.

7. It should be discussed in the main text how food calorie content (absolute and in %) differs under diets with different concentrations of methionine.

Response: The control holidic medium with 40% amino acids contains 95.36 kcal/L, and MetR does 94.49 kcal/L. We maintain that this <1% difference (which is anyway a calculated value from the theoretical energy contributions of the dietary components)

between the foods is insufficient to explain the lifespan differences. We have added the following sentences in the Results.

(Results p.5) This resulted in a negligible (less than 1%) change in the total energy content of the diet from 95.36 kcal/L for the control, to 94.49 kcal/L for MetR.

Reviewer #2 (Remarks to the Author):

The authors have addressed several comments of mine in the revised manuscript, such as the general aging effects of MsrA-null mutation and the differential impacts of methionine-associated metabolites. However, I still have concerns about the mechanisms of MetR. They failed to figure out the major target tissues for “methionine memory” or, alternatively, the biological processes that are regulated by “methionine memory” to modulate aging process (they excluded bmm/Lsd2-associated lipid metabolism though). This work did not provide sufficient significances or novelties into the field without these mechanisms as compared to previously-published studies such as MetR and mitophagy (PMID: 31850341), cell turnover (33902813), glucose sensing (32821821), and so on. I am not convinced by the results in Fig. S5 as well. Therefore, I still recommend the authors to address the following issues:

Response: We appreciate the reviewer for discussion on the novelty of our manuscript. We do not think that our study lacks sufficient significance. The main theme of our study is the age-dependency of the organismal response to MetR, which has not been previously demonstrated in the literature. We have used a multi-omics approach to describe the response in detail at a level not achieved in many other MetR studies. It is well known that MetR induces various beneficial programmes, including decreased anabolism (e.g. translation) and increased catabolism (e.g. autophagy), and we have shown in this study that MsrA is likely to be one of the effectors of MetR-longevity. It is not possible for a single study to test all possible mechanisms, but our data establish a new hypothesis for the mechanistic underpinnings of MetR-longevity that can also be tested in yeast, worms, flies, and rodents.

The reviewer compared our study to three previous ones. These studies are interesting because they show how cells respond to methionine in the medium or in the diet. However, they are of very different context and therefore none of them can be directly comparable to ours. Here is the brief description what each study showed.

- 1, PMID: 31850341, This study performed several genetic experiments to understand MetR-longevity in Yeast. The authors focused on autophagy for the

possible mechanism and found that mitophagy is key for extension of chronological Lifespan by MetR.

2, PMID: 33902813, This study addresses how cell death and proliferation are regulated by oncogenic Src in the imaginal discs of *Drosophila* larvae. The authors found that dietary methionine levels influenced cell proliferation but not cell death. This study does not pertain to ageing.

3, PMID: 32821821, The authors of the study investigated the mechanism of extension of replicative lifespan of yeast by glucose restriction (GR). They showed that GR decreased methionine biosynthesis and uptake, and supplementation of methionine canceled the GR-longevity.

In principle, none of the three studies is related to MetR-longevity in multicellular organisms, which involves tissue/organ specific mechanisms. Two studies (1 and 3) are from yeast. Yeast is useful for genetic analysis to dissect epistatic relationships between two or more genes and pathways for cellular senescence, but does not necessarily delineate biological mechanisms of ageing in multicellular organisms regulated by tissue-interactions and life history. The second study in *Drosophila* is not about lifespan but instead is about tumor growth. Imaginal discs in *Drosophila* larvae provide powerful experimental system to perform cellular and molecular study of development and growth. The second study showed that Met/SAM metabolism cell autonomously contributes to proliferation of Src tumor cells. In contrast, ageing is a complex organismal phenomenon and the effects of Met and its metabolites on ageing can be systemic. Thus, regarding novelty, these three papers cannot be compared with our study.

Given organismal longevity is regulated by multi-organ networks, there must be additive and synergistic effects of various signals from various organs. In addition, it is reasonable to assume that inhibiting *MsrA* induction in one tissue would not alleviate the overall reduction in MetSO and its effects on other tissues, as metabolites can be circulated. Indeed, we found that the levels of MetSO in hemolymph are four times higher than that of Met (new Fig. S3a). Therefore, determination of a responsible tissue for *MsrA*-longevity in the context of dietary MetR (systemic reduction of Met) may not be feasible, especially when an efficient RNAi line and highly tissue-specific gene switch drivers are not available. In line with this, we would like to highlight the

fact that even *daf-2*, the most famous gene for its anti-longevity function discovered 30 years ago, is still under investigation as to where and how it acts in *C. elegans* for longevity (Zhang et al., Nat Commun, 2022). The authors of this study needed to create the auxin-induced protein degradation (AID) tagged *daf2* knock-in allele and tissue-specific TIR-1 (F-Box protein) lines to perform tissue-specific Daf2 degradation.

We do however agree with the reviewer that it would be interesting to discuss where the MetR-*MsrA* axis works to increase lifespan. Therefore, we propose to add the new data of hemolymph MetSO levels in Fig. S3a and a paragraph in the discussion to acknowledge this point.

Supplementary Fig. 3a, Quantification of Met and MetSO in the hemolymph of female Canton-S flies that were fed with a standard yeast-based diet for four days post-eclosion. $n = 3$. For the statistics, a two-tailed Student's *t* test was used.

(Results p.11) In physiological conditions, free-MetSO exists in hemolymph at a level four times more than Met (Supplementary Fig. 3a). This implies that Met is oxidized in body fluids, or MetSO formed in tissues is circulated throughout the body.

(Discussion p.20) This study did not identify the tissue where 1) Met is sensed and 2) *MsrA* functions to drive a pro-longevity mechanism. Previous research using genetic MetR models in *Drosophila* has shown that lifespan extension occurs when Met levels

are decreased either in the gut or in the fat body (Parkhitko et al., PNAS 118, e2110387118, 2021). Their data suggest that MetR in at least these two tissues contribute to the organismal longevity. Another report indicates that overexpression of *gnmt* in the fat body has been found to extend lifespan (Tain et al., Aging, Cell, 2019). These results suggest that a decrease in Met/SAM levels in the gut or fat body can be a key factor in lifespan extension. This is consistent with the fact that we found *MsrA* to be induced in these organs during MetR. However, we also found that Met/SAM/MetSO are metabolites that can be transported through circulation (as they exist in hemolymph). Thus, changes in *MsrA* expression in multiple peripheral tissues, not only in a specific tissue, may be important to decrease systemic MetSO levels to an extent that extends lifespan.

1. They checked knockdown efficiencies of different RNAi lines using Da-GS in Fig. S5A. Does Da-GS-induced *MsrA* knockdown phenocopy *MsrA*-null mutation regarding MetR and lifespan extension? The observations were all obtained from *MsrA*-null mutation, the single genetic evidence in this study.

Response: In Fig. S5A, we utilised da-Gal4 not Da-GeneSwitch, as da-Gal4 is a stronger driver in our experimental setup. However, even with da-Gal4, these RNAi lines were unable to completely block the reduction of MetSO during MetR. Therefore, it is not feasible to examine the effect of *MsrA* knockdown on lifespan extension using these available fly lines.

We only used a null allele of *MsrA* in this study. Nevertheless, the experiments were conducted in a well-controlled genetic background, and careful metabolic and physiological phenotyping supports the crucial role of MetSO. Given the well-documented pro-longevity function of *MsrA* in previous studies, we believe that our data adequately support the requirement of *MsrA* for MetR-longevity.

2. In Fig. S5, MetR extended median lifespan of elav-GS control by ~50%, why only 5% extension in TI-GS and 10% in WB-FB-GS were found? They should repeat these experiments or confirm them ESPECIALLY in the GUT using other GS or Gal4-Gal80TS system. Their results have indicated gut *MsrA* as a critical regulator. It could be the

global effects, not in single organs, of MsrA, but the authors need to validate with solid results.

Response: We appreciate the reviewer's comments and would like to clarify the interpretation of our results. The different magnitudes of lifespan extension seen in the different Gal4/GS drivers likely reflect the varying expression domains, strengths of leaky expression, and genetic backgrounds of the drivers. We acknowledge the reviewer's point that these data are less controlled than our other experiments, and are therefore subject to more variability. In light of this, we have removed these data from the revised manuscript. Our main conclusion regarding the age-dependency of the organismal nutritional response remains valid, and further analysis of tissue specific function of MsrA is beyond the scope of the present study. Once again, we appreciate the reviewer's constructive comments, which led us to add new data and a paragraph in the Discussion that have significantly improved the revised manuscript.

REVIEWER COMMENTS

Reviewer #1 (Remarks to the Author):

The authors have addressed all my comments. Great work!

Reviewer #2 (Remarks to the Author):

I understand the authors might not think the results of MrsA RNAi in whole body and different tissues were good enough, so that they removed them in the current manuscript. However, removing these results actually potentially diminished the significance, quality, as well as novelty, of this wonderful work. As I have mentioned in the previous two revisions, many of the results in the current manuscript have been previously shown or implied in the following studies.

1. MetR extends longevity (PMID: 24710037, 34588310);
2. FoxO regulates MrsA transcription (20655917).
3. Overexpression of MrsA, which is supposed to decrease MetSO, in either whole body or pan-neuron, extends longevity (11867705, 20655917);
4. Possible mechanisms include FoxO activation in a feed-back loop (20655917)

In addition to early-time MetR, exploiting the key tissues responding to MetR or MetSO, or alternatively the molecular mechanisms would remarkably strengthen their conclusion. Otherwise, the RNAseq results in both gut and fat body as well as single-cell RNAseq results in the gut only indicated the correlation between molecular changes and phenotypes.

It is practical and feasible for them to perform MrsA RNAi in different organs to figuring out the important one(s), like gut or fat body. It is also very important evidence to confirm their results from a single MrsA-null mutation and avoid potential genomic perturbation. In the previous revision, MetR only extended lifespan by <5% in MrsA-gut-KD and control flies. MetR greatly, however, extended lifespans (20%-30%) in bmm- and Lsd-gut-KD and control flies. Obviously, the experiments were not controlled very well.

On the other hand, it would not be difficult for them to perform RNAi screening against differentially expressed genes from their RNAseq or single-cell RNAseq data to identify the key genes and the

associated biological activities in the gut, fat body, or even whole body. Then the molecular mechanisms would be sufficiently fulfilled.

Unfortunately, the authors retreated from the two directions. It is really very difficult for me to make the decision of rejection after so many efforts they have made and such great omic results they have obtained. I will leave the final decision to the editor.

Reviewer #2 (Remarks to the Author):

I understand the authors might not think the results of MrsA RNAi in whole body and different tissues were good enough, so that they removed them in the current manuscript. However, removing these results actually potentially diminished the significance, quality, as well as novelty, of this wonderful work. As I have mentioned in the previous two revisions, many of the results in the current manuscript have been previously shown or implied in the following studies.

1. MetR extends longevity (PMID: 24710037, 34588310);
2. FoxO regulates MrsA transcription (20655917).
3. Overexpression of MrsA, which is supposed to decrease MetSO, in either whole body or pan-neuron, extends longevity (11867705, 20655917);
4. Possible mechanisms include FoxO activation in a feed-back loop (20655917)

Response: We thank the reviewer for acknowledging our study by referring "*this wonderful work*" on top of the criticisms. The four points that the reviewer raised here have been honestly and fairly discussed in the manuscript. We believe that the key take-home message of this study is the age-dependency of the organismal response to MetR. Thus, investigating tissue specificity of the MsrA's function would not be mandatory for the present study and we would like to see it as an important direction for the future.

Nevertheless, we agree that we need to discuss tissue specificity. In this 3rd revision, we tested whether the MsrA induction is ubiquitous or specific to some tissues we discussed in the previous manuscript. We therefore performed a qPCR analysis of MsrA in each of five isolated tissues; the brain, the thorax (muscles enriched), the ovary, the gut, and the abdomen (the fat body). The new Supplementary Fig. 3f showed that MsrA is induced by MetR in many tissues but not in the ovary. This suggests that the MsrA induction by MetR is a general phenomenon in many cells, if not all. Given that the tendency of induction in the gut and the fat body is stronger than the other samples, these tissues may contribute more to the longevity phenotypes. We added these data and discussion in the text.

Supplementary Fig. 3f, Quantitative RT-PCR analysis of *MsrA* expression in various tissues of female Canton-S flies fed with or without a methionine-restricted diet for three days. n = 4. For the statistics, one-way ANOVA with Holm-Šidák's multiple comparison test was used.

(Results) *MsrA* transcription is upregulated also in other tissues such as the brain or the thorax (muscle enriched), but not in the ovary (Supplementary Fig. 3f), suggesting that the *MsrA* induction by MetR is a general phenomenon in many cells, if not all.

In addition to early-time MetR, exploiting the key tissues responding to MetR or MetSO, or alternatively the molecular mechanisms would remarkably strengthen their conclusion. Otherwise, the RNAseq results in both gut and fat body as well as single-cell RNAseq results in the gut only indicated the correlation between molecular changes and phenotypes.

It is practical and feasible for them to perform MrsA RNAi in different organs to figuring out the important one(s), like gut or fat body. It is also very important evidence to confirm their results from a single MrsA-null mutation and avoid potential genomic perturbation. In the previous revision, MetR only extended lifespan by <5% in MrsA-gut-KD and control flies. MetR greatly, however, extended lifespans (20%-30%) in *bmm*- and *Lsd*-gut-KD and control flies. Obviously, the experiments were not controlled very well.

On the other hand, it would not be difficult for them to perform RNAi screening against differentially expressed genes from their RNAseq or single-cell RNAseq data to identify the key genes and the associated biological activities in the gut, fat body, or even whole body. Then the molecular mechanisms would be sufficiently fulfilled. Unfortunately, the authors retreated from the two directions. It is really very difficult for me to make the decision of rejection after so many efforts they have made and such great omic results they have obtained. I will leave the final decision to the editor.

Response: We appreciate the reviewer for all the comments above to improve our study. We understand that the reviewer considers our study is not perfect as it lacks tissue specific manipulations and confirmation of the mutant phenotype by RNAi. As we explained in our previous revisions, unfortunately it is difficult to pursue this direction since an effective *MsrA*-RNAi line is not available.

The reviewer allowed us to follow alternative paths to understand the mechanisms for the MetR-longevity, which were informed by our omics data. During our initial round of revision, we tested whether altered lipid metabolism would be relevant to the MetR-longevity. However, the null mutation of *bmm* does not abrogate the MetR-dependent lifespan extension (previous Fig. 6h, now in Supplementary Fig. 5c).

In addition to changes to lipid metabolism, we noticed that several genes involved in autophagy were upregulated in our RNAseq analysis of the young gut (Supplementary Fig. 2). We have now also undertaken experimental work to block

autophagy, using *Atg8a*[KG07569], which is a viable mutant that we backcrossed eight generations into our control genetic background w^{Dah} . *Atg8a*[KG07569] is a protein null mutant with defective autophagy (Nezis *et al.*, *J Cell Biol*, 2008). Strikingly, the lifespan extension by MetR was completely abolished in this mutant (new Supplementary Fig. 6a), suggesting functional autophagy is required for the MetR-longevity. We also found that *MsrA* induction in the gut is mitigated in the *Atg8a* mutant (Supplementary Fig. 6b). Although the mechanism for how defective autophagy influences *MsrA* transcription is not known, it is possible that *MsrA* is a downstream target of autophagy or that *MsrA* and autophagy interact to promote lifespan. In either case, our new observations now confirmed that autophagy is a key mechanism for MetR-induced longevity. Recently, the lifespan extension by early-life rapamycin treatment in *Drosophila* is reported to be dependent on intestinal autophagy (Juricic *et al.*, *Nat aging*, 2022). Although our analysis did not test directly whether autophagy in the gut but not in the other tissue is necessary, MetR in autophagy-defective yeasts has been shown to fail to extend their lifespan (Ruckenstuhl *et al.*, *PloS Genet*, 2014). Therefore, our analysis provided evidence that autophagy promotes lifespan upon MetR and *MsrA* is a possible molecular target of autophagy. These substantial additions of new data are now in the revised manuscript.

Supplementary Fig. 6a, Lifespans of female w^{Dah} and *Atg8a*^{KG07569} flies fed with or without a methionine-restricted diet. Sample sizes (n) are shown in the figure. For the statistics, a log-rank test was used.

Supplementary Fig. 6b, Quantitative RT-PCR analysis of *MsrA* expression in female guts of *w^{Dah}* and *Atg8a^{KG07569}* flies fed with or without a methionine-restricted diet for three days. n = 6. For the statistics, one-way ANOVA with Holm-Šidák's multiple comparison test was used.

(Results) In addition to lipid metabolism, several genes involved in autophagy were upregulated in our RNAseq analysis of the young gut (Supplementary Fig. 2). To test the functional contribution of autophagy to lifespan extension by MetR, we used a *Atg8a^{KG07569}* viable mutant after backcrossing it for eight generations to *w^{Dah}*. *Atg8a^{KG07569}* is a protein null mutant with defective autophagy⁵⁵. Strikingly, lifespan extension by MetR was completely abolished in this mutant, in fact they were significantly shorter lived than flies on the control diet (Supplementary Fig. 6a). We also found that *MsrA* induction in the gut was abolished in the *Atg8a* mutant (Supplementary Fig. 6b), implying that *MsrA* is a downstream target of autophagy, although the mechanism for how autophagy influences *MsrA* transcription is not known. Given that the lifespan extension by early-life rapamycin treatment in *Drosophila* is reported to be dependent on intestinal autophagy¹³ and that MetR in autophagy-defective yeasts has been shown to fail to extend their lifespan⁵⁶, we propose that intestinal autophagy could be a key mechanism for MetR-induced longevity program.

(Discussion) Although we did not quantify activation of these pathways, our data suggest that autophagy is required for the full induction of *MsrA* and lifespan extension by the MetR diet. Considering that *MsrA* is induced downstream of autophagy and that *MsrA* is required for MetR-induced lifespan extension, one of the roles of autophagy during MetR may be to maintain Met levels by inducing *MsrA*. Further study is necessary to understand how autophagy is affected by the diet in an age-dependent manner and how the nutrient-sensing pathways contribute to regulate autophagy in the gut as well as in the other tissues in response to the MetR diet.

Overall, in the three rounds of revision, we have endeavored to improve the study with all possible experimental data we could obtain. Our study would not be incomplete, and it still has many unanswered questions, but we are confident that our manuscript provides significant advancement of the understanding of the mechanisms MetR-longevity and its age-dependency. These data and the accompanying resources we have created, will be of interest to a broad readership.

REVIEWERS' COMMENTS

Reviewer #2 (Remarks to the Author):

The authors did not fully confirm the organ-specific effects of early metabolic restriction (MetR), but I understand the challenges of this type of study (RNAi efficiency and so on). The authors explained the technical difficulties and other fundamental issues in the field. They have also included autophagy-associated impacts that could explain the mechanism, even though autophagy regulation has been reported in other lifespan-extending strategies. I would like to recommend the acceptance.

Reviewer #2 (Remarks to the Author):

The authors did not fully confirm the organ-specific effects of early metabolic restriction (MetR), but I understand the challenges of this type of study (RNAi efficiency and so on). The authors explained the technical difficulties and other fundamental issues in the field. They have also included autophagy-associated impacts that could explain the mechanism, even though autophagy regulation has been reported in other lifespan-extending strategies. I would like to recommend the acceptance.

>Response: We express profound gratitude to the reviewer for recommending the acceptance of our revised manuscript, although the study is not assessing the organ-specific effects. While we are pleased with this comment, we should discuss an issue that has recently come to our attention.

After submitting the revised manuscript, we discerned an anomaly with the autophagy mutant *Atg8a*^{KG07569}, which we employed for the revision. As per the Flybase records, this mutant has a P-element insertion in its first exon, leading to a loss of function allele. Contrarily, our observations noted the unaltered expression of *Atg8a* mRNA (qPCR) and protein (western blotting) within this mutant. Careful PCR analysis revealed that our in-house stock curiously lacked this insertion, though it retained the w+ marker in its genome, a marker we used to ascertain the presence of the P-element insertion when backcrossing. We postulate that either during the backcrossing or its subsequent stock maintenance, the P-element insertion might have gone from the original site, but there might remain a potential secondary insertion. Given that we cannot guarantee the presence of the insertion at the original *Atg8a* locus when we performed the experiment for the revision, it would be imprudent to rely on data sourced from this mutant's analysis. Nevertheless, our assessment of the backcrossed mutant stock distinctly showed an indication of autophagy defect, as evidenced by the accumulation of p62/Ref(2)P (Figure R1). While this suggests that our mutant stock is indeed an autophagy defective, the nature of this mutant remains enigmatic. Given this complicated situation, we have taken the judicious decision to exclude these data from the manuscript.

We agree with the reviewer that autophagy has been already shown to be required for many lifespan-extending interventions, including MetR at least in yeast (Ruckenstuhl et al., PloS Genet, 2014). The crux—and thereby the novelty—of our study is to highlight the age dependency of dietary responses and pivotal role of MsrA. Excising

this supplementary figure showing the involvement of autophagy should not diminish the impact of the study. In our RNAseq analysis of the young gut upon Met depletion, we noticed an upregulation of several genes related to autophagy (Supplementary Fig. 2). Moreover, MetR lowered p62 levels, suggesting that MetR might enhance autophagy (Fig. R2). We found FoxO activation during MetR, which, according to previous studies (Juhasz et al., Cell Death Diff, 2007; Demontis and Perrimon, Cell, 2010), can potentially trigger autophagy. A thorough examination is vital to understand the dynamics and mechanisms behind MetR's influence on autophagy, how autophagy malfunctions in aged organisms, the lasting effects after stopping the MetR diet, and the specific tissues critical for lifespan extension. Furthermore, it is compelling to explore the mechanistic understanding of the interplay among FoxO, MsrA, and autophagy in the context of lifespan extension. With the removal of Supplementary Figure 6, we would like to discuss these relationships in our paper. However, delving deep into such intricate realms by experiments would necessitate an extended timeline. Thus, it transcends the purview of our current manuscript, and we anticipate earmarking it as a focal point of our ensuing research endeavours.

Fig. R1, Western blot analysis of p62/Ref(2)P in the *Atg8a*[KG07569] mutant. YB indicates yeast-based diet. Both males and females show an accumulation of p62, an indicative of defective autophagy.

Fig. R2, Western blot analysis of p62/Ref(2)P in the MetR condition.

We removed Supplementary Fig.6 and the following paragraphs from the text.

~~(Results) In addition to lipid metabolism, several genes involved in autophagy were upregulated in our RNAseq analysis of the young gut (Supplementary Fig. 2). To test the functional contribution of autophagy to lifespan extension by MetR, we used a *Atg8a^{KG07569}* viable mutant after backcrossing it for eight generations to *w^{Dah}*. *Atg8a^{KG07569}* is a protein null mutant with defective autophagy⁵⁵. Strikingly, lifespan extension by MetR was completely abolished in this mutant, in fact they were significantly shorter lived than flies on the control diet (Supplementary Fig. 6a). We also found that *MsrA* induction in the gut was abolished in the *Atg8a* mutant (Supplementary Fig. 6b), implying that *MsrA* is a downstream target of autophagy, although the mechanism for how autophagy influences *MsrA* transcription is not known. Given that the lifespan extension by early life rapamycin treatment in *Drosophila* is reported to be dependent on intestinal autophagy¹³ and that MetR in autophagy-defective yeasts has been shown to fail to extend their lifespan⁵⁶, we propose that intestinal autophagy could be a key mechanism for MetR-induced longevity program.~~

~~(Discussion) Although we did not quantify activation of these pathways, our data suggest that autophagy is required for the full induction of *MsrA* and lifespan extension by the MetR diet. Considering that *MsrA* is induced downstream of autophagy and that *MsrA* is required for MetR-induced lifespan extension, one of the roles of autophagy during MetR may be to maintain Met levels by inducing *MsrA*. Further study is necessary to understand how autophagy is affected by the diet in an age-dependent manner and how the nutrient-sensing pathways contribute to regulate autophagy in the gut as well as in the other tissues in response to the MetR diet.~~

We added the following sentences in the Discussion.

In our RNAseq analysis of the young gut upon Met depletion, we noticed an upregulation of several genes related to autophagy (Supplementary Fig. 2), which might be attributable to the FoxO activation^{35,66}. Given that the lifespan extension by early-life rapamycin treatment in *Drosophila* is reported to be dependent on intestinal autophagy¹³ and that MetR in autophagy-defective yeasts has been shown to fail to extend their lifespan⁶⁷, upregulation of autophagy in the gut or other tissues could be a key mechanism for MetR-induced longevity program. It is compelling to explore the mechanisms of the interplay among FoxO, MsrA, and autophagy in the context of lifespan extension by early MetR.

REVIEWERS' COMMENTS

Reviewer #2 (Remarks to the Author):

I apologize for the delayed responses, but I was hesitant about making a final decision. It would certainly be ideal to see the molecular mechanisms by which early MetR extends lifespan through autophagy regulation. However, I appreciate the authors' honesty in reporting the defects of the autophagy lines they used and their request to withdraw certain important data. I have carefully reviewed the revised manuscript, which includes a comprehensive discussion of the findings of autophagy regulation in previous research, the RNAseq results and other findings associated with autophagy in the current study, as well as future directions for validating the mechanisms. I believe it is suitable to accept the manuscript if the responses to all reviewers' comments are included as supplementary files.

Reviewer #2 (Remarks to the Author):

I apologize for the delayed responses, but I was hesitant about making a final decision. It would certainly be ideal to see the molecular mechanisms by which early MetR extends lifespan through autophagy regulation. However, I appreciate the authors' honesty in reporting the defects of the autophagy lines they used and their request to withdraw certain important data. I have carefully reviewed the revised manuscript, which includes a comprehensive discussion of the findings of autophagy regulation in previous research, the RNAseq results and other findings associated with autophagy in the current study, as well as future directions for validating the mechanisms. I believe it is suitable to accept the manuscript if the responses to all reviewers' comments are included as supplementary files.

>Response: We sincerely appreciate the reviewer's thoughtful comments on our revised manuscript. Ensuring the accuracy and transparency of our research is paramount to us. In line with the reviewer's suggestion, we will include responses to all comments from reviewers as supplementary files to bolster clarity and comprehensiveness. We are pleased that the reviewer found the revised manuscript comprehensive, especially in relation to the discussion of autophagy regulation and the outlined future directions. We would like to extend our gratitude once again for the invaluable feedback from the reviewer, which has significantly improved the study.